

# Ozone profiles by DIAL at Maïdo Observatory (Reunion Island) Part 1. Tropospheric ozone lidar: system description, performances evaluation and comparison with ancillary data

Valentin Duflot[1,2], Jean-Luc Baray[3], Guillaume Payen[2], Nicolas Marquestaut[2], Francoise Posny[1], Jean-Marc Metzger[2], Bavo Langerock[4], Corinne Vigouroux[4], Juliette Hadji-Lazaro[5], Thierry Portafaix[1], Martine De Mazière[4], Pierre-Francois Coheur[6], Cathy Clerbaux[5,6], and Jean-Pierre Cammas[1,2]

[1]Laboratoire de l'Atmosphère et des Cyclones (LACy), UMR8105, Saint-Denis de La Réunion, France
[2]Observatoire des Sciences de l'Univers de La Réunion (OSUR), UMS3365, Saint-Denis de la Réunion, France
[3]Laboratoire de Météorologie Physique (LaMP), UMR6016, Observatoire de Physique du Globe de Clermont-Ferrand, CNRS - Université Blaise Pascal, Clermont-Ferrand, France
[4]Royal Belgian Institute for Space Aeronomy (BIRA-IASB), 3, Av. Circulaire, 1180, Brussels, Belgium
[5]LATMOS/IPSL, UPMC Univ. Paris 06 Sorbonne Universités, UVSQ, CNRS, Paris, France
[6]Spectroscopie de l'Atmosphère, Service de Chimie Quantique et Photophysique, Université Libre de Bruxelles (ULB), Brussels, Belgium

*Correspondence to:* Valentin Duflot (valentin.duflot@univ-reunion.fr)

**Abstract.** Recognizing the importance of ozone in the troposphere and lower stratosphere in the tropics, a DIAL tropospheric ozone lidar system (LIO3T$_{UR}$) was developed and installed at the Université de la Réunion campus site (close to the sea) in Reunion Island (southern tropics) in 1998. From 1998 to 2010, it acquired 427 ozone profiles from the low to the upper troposphere and has been central to several studies. In 2012, the system was moved up to the new Maïdo Observatory facility

5    (2160m above mean sea level - amsl) where it started operation in February 2013. The current system (LIO3T) configuration generates a 266nm beam obtained with the fourth harmonic of a Nd:YAG laser sent into a Raman cell filled up with deuterium (using helium as buffer gas) generating the 289 and 316nm beams enabling the use of the DIAL method for ozone profile measurements. Optimal range for the actual system is 6-19km amsl, depending on the instrumental and atmospheric conditions; for a 1-hour integration time, vertical resolution varies from 0.7km at 6km amsl to 1.3km at 19km amsl, and mean uncertainty

10    within the 6-19km range is between 6 and 13%. Comparisons with 8 electrochemical concentration cell (ECC) sondes simultaneously launched from the Maïdo Observatory show a good agreement between datasets with a 7.7% mean absolute value of the relative differences with respect to the mean (*D*) between 6 and 17km amsl (LIO3T low); comparisons with 37 ECC sondes launched from the nearby Gillot site during day time in a ± 24-hour window around lidar shooting result in a 10.3% *D* between 6 and 19km amsl (LIO3T low); comparisons with 11 ground-based Network for Detection of Atmosphere Composition

15    Change (NDACC) Fourier Transform Infrared (FTIR) spectrometer measurements acquired during day time in a ± 24-hour window around lidar shooting show a good agreement between datasets with a *D* of 11.8% for the 8.5-16km partial column (LIO3T high); and comparisons with 39 simultaneous Infrared Atmospheric Sounding Interferometer (IASI) observations over Reunion Island show a good agreement between datasets with a *D* of 11.3% for the 6-16km partial column (LIO3T high). ECC, LIO3T$_{UR}$ and LIO3T O$_3$ monthly climatologies all exhibit the same range of values and patterns. In particular, the southern




hemisphere biomass burning seasonal enhancement, the ozonopause altitude decrease in late austral winter-spring, as well as the signature of deep convection bringing boundary layer-ozone poor air masses up to the mid-upper troposphere in late austral summer, are clearly visible on all datasets.

# 1 Introduction

Ozone is a major greenhouse gas in the upper troposphere and lower stratosphere (Lacis et al., 1990), an important pollutant, and impacts the oxidative capacity of the atmosphere (Martin et al., 2003). In the troposphere, the ozone budget is influenced by transport from the stratosphere, by in situ photochemical production associated with ozone precursors emitted by anthropogenic activity, biomass burning, lightning and by surface deposition (Stevenson et al., 2006).

Reunion Island is a tropical island located in the south-western part of the Indian Ocean at 20.8°S and 55.5°E. It is season-
ally impacted by biomass burning plumes transported from Southern Africa, South America and South-East Asia which can significantly affect the free tropospheric concentrations of ozone and other pollutants like CO (Edwards et al., 2006 ; Duflot et al., 2010). Moreover, it is affected by stratospheric intrusions associated with the dynamical influence of the subtropical jet stream (Baray et al., 1998; Clain et al., 2010) and the tropical cyclone deep convection (Leclair de Bellevue et al., 2006).

The barrier effect and dynamical exchanges between the tropical reservoir and midlatitudes, and vertically between the
troposphere and the stratosphere are of great interest to document climate change. Tropospheric ozone measurements are then performed routinely in Reunion Island by ozone sondes at the Gillot site (20.893°S, 55.529°E, 9m above mean sea level (amsl) ; cf. Figure 1) since 1992 (in the framework of the Network for the Detection of Atmospheric Composition Change - NDACC since 1996 and of Southern Hemisphere ADditionnal OZone sondes - SHADOZ network since 1998), and by lidar at the Université de la Réunion campus site (20.902°S, 55.485°E, 80m amsl) since 1998 (Baray et al., 2006).

To improve the operation of remote sensing instruments, a 2160m-high atmospheric facility was built in 2012 at the summit of the Maïdo mount (21.079°S, 55.383°E, 2160m amsl - cf. Figure 1), and most of the instruments previously installed close to the coast at the Université de la Réunion campus site were moved up to this new facility along the year 2012 (Baray et al., 2013). Being inside the boundary layer during the day and most of the time inside the free troposphere during the night (except during the warm and rainy season), the Maïdo Observatory is dedicated to the investigation of the boundary layer composition
and processes (especially in the framework of the Global Atmospheric Watch network - GAW), as well as to the study of the low-middle atmosphere (especially in the framework of the NDACC). Four lidar systems are permanently deployed and routinely operated at the Maïdo Observatory:

   - a Doppler wind lidar dedicated to the study of the middle atmosphere dynamics (Khaykin et al., 2015),

   - the LIO3S, a lidar dedicated to stratospheric ozone measurements (Portafaix et al., 2003; Portafaix et al., 2015),
- the LI1200, a lidar dedicated to tropospheric water vapor (Hoareau et al., 2012; Dionisi et al., 2015; Vérèmes et al., 2015) and stratospheric-mesospheric temperature measurements (Morel et al., 2002; Keckhut et al., 2004; Sivakumar et al., 2011a),





- and the LIO3T lidar (Baray et al., 1999; Keckhut et al., 2004; Baray et al., 2006; Clain et al., 2009, 2010; Vérèmes et al., 2016) dedicated to the observation of tropospheric ozone (as well as aerosols from the free troposphere up to the lower stratosphere).

It is noteworthy that the LIO3T system was very recently affiliated in the NDACC for ozone measurements ; this paper aims
to provide a technical reference socle for further use of the ozone data provided by the LIO3T system: we first give a brief historical review of the tropospheric ozone lidar system and measurements performed when installed at the Université de la Réunion campus site (1998-2010), we then describe the current LIO3T system installed at the Maïdo Observatory. We show comparisons between the LIO3T ozone measurements and ancillary data, and we finally present an overview of the 2013-2015 Maïdo Observatory ozone measurements database.

In the following, the system will be referred as "LIO3T$_{UR}$" when installed at the Université de la Réunion (for LIdar O$_3$ Troposphéric Université de la Réunion), and the current system (installed at the Maïdo Observatory) will be referred as "LIO3T".

## 2   Brief historical review of the lidar system at the Université de la Réunion campus site (1998-2010)

This section aims to provide a review of the LIO3T$_{UR}$ system and resulting dataset. It gives an overview of the historical context
preceding the operation of the tropospheric ozone lidar at the Maïdo Observatory. It also presents the main characteristics of the LIO3T$_{UR}$ system, its performances and database, and shows a compilation of the resulting observations over the 13 years of operation from 1998 to 2010.

### 2.1   Instrumental set up and technical choices

A Rayleigh-Mie lidar was first installed at the Université de la Réunion campus site in 1993 to monitor stratospheric and
mesospheric aerosols in the southern tropics. From 1993 to 1998, the lidar system evolved both in terms of emission and reception (Nd:YAG laser replacement, mosaic telescopes addition, polarization channels installation, infrared channel reception set up) to improve aerosols detection and characterization, and to allow stratospheric-mesospheric temperature measurement.

In 1998, a tropospheric ozone extension was installed to the existing system (Baray et al., 1999). The goal was to perform ozone measurements over the entire tropospheric column, including the upper troposphere. To achieve this, the 289 and 316nm
wavelengths pair was chosen because it affords a good trade-off among the needs for sensitivity in both free troposphere and tropopause region and sufficient laser energy at the implied Raman Stokes lines. The 266nm beam obtained with the fourth harmonic of the Nd:YAG laser was therefore sent into a Raman cell filled up with deuterium generating the 289 (1st Stokes) and 316nm (2nd Stokes) beams enabling the use of the DIAL method for ozone profile measurements. After collection through telescopes and optical fibers, the 289 and 316nm signals were spectrally separated with a spectrometer that includes a Czerny-
Turner holographic grating and sent toward Hamamatsu R1527P PMTs. An analog channel was used for the lower layers and a photoncounting channel for the mid- and upper troposphere. Baray et al. (1999) give a complete description of the 1998 system





and provide justifications of the technical choices that were made at this time. Note that the first "home made" acquisition chain was changed for a LICEL one in 2007, but this change did not cause significant differences in the profiles acquired.

The description of the well-known DIAL retrieval scheme of the ozone profile from the backscattered ON (absorbed by ozone, here: 289nm) and OFF (less absorbed by ozone, here: 316nm) signals can be found in, e. g., Pelon and Mégie (1982)
and Mégie et al. (1985). Molina and Molina (1986) ozone cross sections were used for ozone profile retrieval.

## 2.2 Performances, dataset and ozone climatology

The LIO3T$_{UR}$ optimal range was 3.5-17km amsl. Figure 2 gives the mean uncertainty and resolution over the 13 years of operation. The mean uncertainty varies from 6% at 3km amsl to almost 60% at 17km amsl. Vertical resolution goes from 0.1km at 3km amsl to 1.8km at 17km amsl. The temporal resolution (or integration time) was chosen depending on the atmospheric
conditions and varied roughly between 40 minutes and 3 hours. Figure 3 shows the number of acquired profiles per year from 1998 to 2010. 427 tropospheric ozone profiles were acquired over the 13-year operation period. Measurements were stopped in 2010 to prepare the move up to the Maïdo Observatory.

Figure 4 (top right panel) shows the resulting monthly tropospheric ozone climatology. For comparison, the monthly tropospheric ozone climatology derived from the electrochemical concentration cell (ECC) sondes measurements performed at the
Gillot site (Figure 1) between 2005 and 2015 is shown (top left panel). Note that for comparison convenience, whereas Maïdo Observatory tropospheric ozone system, retrieval, performances and database descriptions are the subject of the following sections, the tropospheric ozone monthly climatology resulting from the lidar observations at the Maïdo Observatory is also plotted on Figure 4 (bottom panel) but will be discussed only in Section 5. We focus here on the description and comparison of the ECC and LIO3T$_{UR}$ climatologies (Figure 4, top panels) in order to estimate the consistency between datasets and to
highlight the main climatological patterns.

The following seasonal features can be observed on both ECC and LIO3T$_{UR}$ climatologies:

- a clear increase of ozone abundance over the whole tropospheric column - especially between 2 and 10 km amsl - starting in June and ending in December with a maximum in October of $\approx 10 \times 10^{11}$ molec/cm$^3$ on average between 4 and 10 km amsl; this increase is due to the influence of air masses coming from South America, Southern Africa and South-East Asia (Edwards
et al., 2006; Duflot et al., 2010) where the biomass burning season occurs every year during this period; ozone abundance then presents a slow decay over the entire tropospheric column from January to May;

- the ozonopause altitude decrease from $\approx$ 17 km in December-July down to $\approx$ 15 km amsl in August-November (Sivakumar et al., 2011b), which is likely a combination of the spring and summer maximum of occurrence of stratosphere-to-troposphere exchanges (STE) above Reunion Island (Clain et al., 2010) and of the winter time thermal effect on the troposphere thickness.

## 3  The current tropospheric ozone system at the Maïdo Observatory (2013-present)

Late 2012, the Maïdo Observatory new facility was complete and many instruments were moved from the Université de la Réunion campus site and installed in the Observatory. Since temperature measurements are now performed with another lidar





system - also dedicated to water vapor measurement (Dionisi et al., 2015; Vérèmes et al., 2015) - the previous LIO3T$_{UR}$ was modified into a system dedicated to the measurement of tropospheric ozone (and aerosols): the "LIO3T".

## 3.1 Description of the current system

We focus here only on the DIAL ozone part of the LIO3T system. Figure 5 sketches the experimental schematic of the ozone DIAL part of the LIO3T and Table 1 gives its main technical characteristics. The LIO3T system mainly relies on the LIO3T$_{UR}$ design (Baray et al., 1999). We use the same approach to generate a 266nm beam going through a deuterium filled Raman cell (using helium as buffer gas) shifting the incoming frequency to 289 and 316nm signals. Operating at a repetition rate of 30 Hz, the 266nm UV pulse energy is currently around 40 mJ per pulse. Backscattered photons are collected by a 4x500mm-telescope mosaic focusing on 1.5mm diameter optical fibers. The optical system allows to obtain an image quality (pulse response of a star image located at the infinite) associated to the atmospheric turbulence of 0.8 mrad. Concerning the part associated to the quality of the receiving telescopes, 90% of the energy is included in 0.6 mrad. After a system of three lenses used to reduce the divergence of the beam by a factor 3, and to adapt the numerical aperture of fibers to that of the grating, the spectral separation of the 289 and 316 nm beams is obtained by a high performance Czerny-Turner holographic grating (3600 lines/mm). Then each beam is redirected towards photomultiplier tubes (PMT) by concave mirrors. Hamamatsu R9980-110 and R7400P-03 photomultipliers tubes are used, for 289 and 316 nm channels, respectively. A Licel time-correlated photon counting unit is used to discriminate both channels energy-shifted photons, triggered by the optical laser signal at 30 Hz with a 150 m resolution.

For information, the detection and characterization of the tropospheric aerosols by the LIO3T system is currently performed using the emitted 532nm "residual" beam, a 200mm telescope for reception of the elastic signal, and a polarization detection system. This aerosols detection wing of the LIO3T system will be the subject of dedicated studies.

## 3.2 Data processing

### 3.2.1 Lidar equation

The lidar DIAL technique (Hinkley, 1976) relies on the difference between two backscattered lidar signals at two different wavelengths, one being relatively strongly (ON, here: 289nm) absorbed by the target species, the other one being relatively weakly absorbed (OFF, here: 316nm). The ozone number density $n_{O3}(z)$ at altitude $z$ (in molec/cm$^3$) is retrieved from the Rayleigh lidar signals according to the following equation:

$$n_{O3}(z) = \frac{-1}{2\Delta\sigma_{O3}(z)} \frac{d}{dz}\left[ln\left(\frac{P(\lambda_{ON},z) - B(\lambda_{ON},z)}{P(\lambda_{OFF},z) - B(\lambda_{OFF},z)}\right)\right] + \delta n_{O3}(z) \tag{1}$$

where $\Delta\sigma_{O3}(z) = \sigma_{O3}(\lambda_{ON},z) - \sigma_{O3}(\lambda_{OFF},z)$ is the differential ozone absorption cross-section at altitude $z$, $P(\lambda_i,z)$ is the number of detected photons at the wavelength $\lambda_i$ and at the altitude $z$, $B(\lambda_i,z)$ is the background noise and detector





noise at the wavelength $\lambda_i$ and at the altitude $z$, and $\delta n_{O3}(z)$ is a correction term corresponding to the absorption by other constituents of the atmosphere, expressed as follows:

$$\delta n_{O3}(z) = \frac{1}{\Delta\sigma_{O3}(z)} \left[ \frac{1}{2}\frac{d}{dz}\left[ ln\left( \frac{\beta(\lambda_{ON},z)}{\beta(\lambda_{OFF},z)} \right) \right] - \Delta\sigma_{atm}(z)n_{atm} - \sum_{ig}\Delta\sigma_{ig}(z)n_{ig}(z) \right] \tag{2}$$

where $\beta(\lambda_i,z)$ is the coefficient of extinction of the molecules and particles at the wavelength $\lambda_i$ and at the altitude $z$, $\Delta\sigma_{atm}(z)$ and $n_{atm}$ the differential cross-section and the density of the atmosphere at the altitude $z$, respectively, and $\Delta\sigma_{ig}(z)$ and $n_{ig}(z)$ the differential cross-section and the number density of interfering gas $ig$ at the altitude $z$, respectively. The background light, the saturation of the detector and the noise from detectors must be added to this equation.

### 3.2.2 Desaturation, filtering and vertical resolution

Saturation can occur only below 7km amsl. To correct it, we apply the scheme described in Pelon (1985, Annex 2). To calculate equations (1) and (2), we apply a derivative filter, which can be expressed as:

$$S_f(k) = \sum_{n=-N}^{N} c_n S(k+n) \tag{3}$$

where $S_f(k)$ is the filtered signal, $S(k)$ is the signal to be filtered and $c_n$ the filter coefficients. These coefficients define the type of filter: low-pass, derivatives, etc. Increasing the number of points of the filter reduces the noise in the signal but degrades the vertical resolution. The calculation of the vertical resolution from the filter parameters is described in Leblanc et al. (2016a). In our case, we use the frequency approach: we calculate the transfer function of the filter and we decide that the frequency for which the gain is lower than 0.5 is the cut-off frequency. We obtain the vertical resolution by dividing the initial resolution by the cut-off frequency. We use the Savitzky-Golay filter of order 2, also called least-squares smoothing filter (Savitzky and Golay, 1964). The resolution rises with the altitude according to a polynomial function (cf. Section 3.3 and Figure 6).

### 3.2.3 Uncertainty

Uncertainties calculation for DIAL ozone retrievals are described in Leblanc et al. (2016b). The most significant sources of uncertainties are found to be the detection noise, the ozone cross section uncertainties and the background noise.

The detection noise can be estimated as a Poisson noise because we use our acquisition card in photo counting mode. The associated error is:

$$\sigma_s(z) = \sqrt{s(z)} \tag{4}$$

where $\sigma_s(z)$ is the error of signal $s$ at the altitude $z$, and $s(z)$ the signal at the altitude $z$. The used ozone cross-sections are defined by Bass and Paur (1984) with an uncertainty equal to 5The background noise includes the background light, which is





altitude-independent, and the detector noise - dark noise and induced signals -, which are altitude-dependent. We use then a linear or polynomial regression to remove the background noise.

To take into account the propagation of these errors in the lidar equation, we use the following equation of propagation:

$$
\begin{cases}
y = f(x_1, x_2, ..., x_N) \\
\sigma_y = \sum_{n=1}^{N} \left( \frac{\partial y}{\partial x_n} \right)^2 u_n^2
\end{cases}
\tag{5}
$$

where $y$ is the signal depending on variable $x_i$, $\sigma_y$ the error of the signal $y$, $\frac{\partial y}{\partial x_n}$ the derivative of $y$ with respect to the variable $x_i$, and $u_i$ the uncertainty of $x_i$. In this equation, we suppose that all uncertainties are independent.

The program used to calculate the ozone profile, uncertainties and resolution is adapted from the stratospheric ozone program "DIAL", which has been described and inter-compared by Godin et al. (1999) and is currently used for the stratospheric DIAL ozone retrievals at Reunion (NDACC affiliated). In this new tropospheric ozone version, cross-sections are adapted to the used
wavelengths, and the uncertainty on these ozone cross-sections is taken into account.

Future plans for the data processing are to: i) use the Brion-Daumont-Malicet (Daumont et al., 1992; Brion et al., 1993; Malicet et al., 1995) ozone cross sections as recommended by the NDACC lidar working group; ii) calculate the uncertainty on the retrieved ozone profile due to the desaturation scheme (which should increase our uncertainty only below 7 km amsl); iii) use the International Space Science Institute (ISSI) team standardized approach for the propagation of the background
and saturation corrections uncertainties (Leblanc et al., 2016b); iv) implement uncertainties calculation due to the presence of aerosols in the troposphere using an iterative aerosol assessment procedure and simultaneous aerosols lidar measurements at 532nm.

### 3.3   Performances

The altitude of the Maïdo Mount being 2160m amsl, the transfer of the tropospheric ozone DIAL system from the University
(80m amsl) to this location increases the upper limit of the profile probed, but also increases the lower limit: the optimal range is now 6-19km (compared to 3.5-17 km range for the LIO3T$_{UR}$). The free troposphere, the tropical tropopause layer (TTL) and lower stratosphere are thus covered by the current system. It is worth mentioning, however, that depending on experimental conditions (lidar alignment, stability of emitted power, laser and Raman cell, atmospheric conditions, etc.), the validity domain can vary from one day to another.
The LIO3T system is only operated at night, twice a week in routine conditions (i.e. out of campaigns). We use three main integration times: 20 minutes for night time series, 1 hour for comparison with collocated ECC soundings, and $\approx$ 3 hours (between $\approx$ 2 and $\approx$ 4 hours, depending on the clear sky time duration) for "full night" profiles. Figure 6 presents examples of resulting profiles and corresponding resolutions for these three integration times for the same night (9th July 2013). Ozone vertical distributions and resolutions differ notably when considering the different integration times, the first one - obviously -
because of the tropospheric ozone variability, the latter because of the choice of the filtering coefficients. For the 20-minutes



integration time, the resulting resolutions are 0.9 and 1.6 km at 6 and 19 km amsl, respectively; for the 1-hour integration time, they are 0.7 and 1.3 km at 6 and 19 km amsl, respectively; and for the 3-hour integration time, 0.4 and 1.2 km.

Figure 7 shows the mean uncertainties for the three main integration times. Unsurprisingly, we find greatest mean uncertainties for the shortest integration time. For the 1- and >1-hour integration times, mean uncertainty goes from 6% at 6 and 19km amsl to 13% at 14km amsl, and for the 20 min integration time, the mean uncertainty goes from 7% at 6km amsl to 16% at 16.5km amsl. These figures are in agreement with the recently published work of Leblanc et al. (2016b) showing uncertainty profiles for DIAL tropospheric ozone measurements between 7 and 11%.

## 4    Comparisons

The goal of this section is to validate the LIO3T ozone measurements by comparing them to ancillary data. Four types of correlative data are used here: collocated ECC soundings (i.e. launched from the Maïdo Observatory during a lidar shooting), routine NDACC/SHADOZ ECC soundings performed during daytime at the Gillot site (cf. Figure 1), and Fourier Transform InfraRed spectrometer (FTIR) tropospheric partial columns measurements from both daytime ground-based and nighttime space-borne instruments.

In the following, we compare $N$ LIO3T ozone measurements $M_{LIO3T}$ with $N$ correlative data $M_{CD}$ by calculating the mean absolute value of the relative differences with respect to the mean $D$ (in %) defined as:

$$D = \frac{200}{N} \sum_{n=1}^{N} \frac{|M_{LIO3T_n} - M_{CD_n}|}{M_{LIO3T_n} + M_{CD_n}} \qquad (6)$$

### 4.1    Comparison with ECC

ECC sondes measure the oxidation of a potassium iodine (KI) solution by $O_3$ (Komhyr et al., 1995). Their accuracy is 5-10% throughout the troposphere and TTL (Smit et al., 2007) and they are commonly used for the validation of ground-based and space-borne ozone observations. Here below, we compare LIO3T ozone profiles with both collocated Maïdo ECC soundings and Gillot SHADOZ/NDACC routine daytime ECC soundings. All these ECC profiles are generated following the "Guidelines for homogenization of ozonesonde data" (Smit et al., 2012). The Gillot SHADOZ/NDACC reprocessed ECC dataset was recently presented by Posny et al. (2016) and Smit et al. (2016). One caveat should be kept in mind: these ECC soundings were performed using the ENSCI/0.5% full buffer solution, instead of the standard half buffer. As of this writing, there is no transfer function to convert these ozone profiles - using non-standard ECC/solution pairing - to the standards. The only certitude is that this ENSCI/0.5% full buffer solution tends to overestimate the amount of ozone by a few percents in the troposphere (Johnson et al., 2002). As a consequence, these ECC profiles should be considered as preliminary data and the $D$ between them and LIO3T profiles stated hereafter is most probably slightly overestimated.

Figure 8 shows the comparison between LIO3T and collocated ECC soundings: two were performed in June 2013, four in May 2015 and two in July 2015. Note that these last six were part of the Maïdo ObservatoRy Gaz and Aerosols Ndacc





Experiment (MORGANE) campaign that took place in May-July 2015 (Portafaix et al., 2015; Vérèmes et al., 2015; Duflot et al., 2016a; Posny et al., 2016). The integration time for the LIO3T profiles used here is 1h and corresponds roughly to the time for the balloon to travel the troposphere. Note that the "discontinuities" in the mean profiles shown on Figure 8 are caused by the varying valid ranges in the LIO3T profiles (cf. Table 2), and note that no profile goes above 17km for these eight

comparisons. In particular, valid range in May and July 2015 (during the MORGANE campaign) is bounded up at 17km by the strong volcanic aerosol loading coming from the Calbuco volcano, which erupted late April 2015 and whose volcanic plume reached the TTL above Reunion Island on the 6th May 2015 before slowly vanishing until the end of July 2015 (Bègue et al., 2016). In the layer where this volcanic plume lies (i.e. between 17 and 22km), we consider that the SO2 and aerosols loading is too strong to allow a correct $O_3$ retrieval (Ancellet et al., 1987; MacGee et al., 1993).

One can see on Figure 8 that there is an overall agreement between LIO3T and the ECC considering the lidar uncertainty and ECC accuracy (right). The mean $D$ is 7.7% for whole probed column (LIO3T low). This value agrees with the ones recently reported for single or multiple ECC-lidar comparisons (between 6 and 20% reported by Uchino et al., 2014; 20% reported by Sullivan et al., 2015; 8% reported by Gaudel et al., 2015).

    Stratosphere-to-troposphere exchanges were observed above Reunion Island in May 2015 during the MORGANE cam-

paign (Duflot et al., 2016a,b). Enhanced aerosol loadings (likely coming from the Calbuco eruptions) were observed in these stratospheric air masses entering the troposphere above Reunion Island, which could have disturbed the ozone detection and quantification by the LIO3T system, and consequently lower the agreement between LIO3T and ECC soundings during this period. Future consideration of the LIO3T simultaneous aerosols measurements in the data processing scheme should help to improve the related uncertainties on the retrieved $O_3$ profiles.

Figure 9 shows the comparison between the SHADOZ/NDACC Gillot routine ECC soundings and LIO3T profiles. As the first ones are performed during daytime (usually around 3PM local time) and the last ones during night time (between 7PM and 1AM local times), ECC soundings are taken into consideration when performed one day before or after a LIO3T profile acquisition; we find 37 pairs for comparison over the years 2013-2015. The LIO3T profiles used here are "full night" profiles. Once again, note that the "discontinuities" in the mean profiles shown on Figure 9 are caused by the varying valid ranges

in the LIO3T profiles (and one can see that only one profile goes above 18km). Despite the fact that the instruments were neither collocated in time nor space (the ECC launch site - Gillot - is ≈ 20km away from the Maïdo Observatory and balloons are advected by the wind), one can see that there is an overall good agreement between measurements considering the lidar uncertainty and ECC accuracy, with a mean $D$ equal to 10.3% over the entire 6-19km column (LIO3T low).

    One should remind that these very satisfactory agreements between ECC and LIO3T profiles should slightly improve in the

future when a transfer function to convert these ECC ozone profiles to the standards is set up.

## 4.2    Comparison with ground-based and space-borne FTIRs

In this section we compare the LIO3T $O_3$ profiles with collocated partial column measurements performed by two FTIRs: the Bruker 125HR installed at the Maïdo Observatory since 2013, and the Infrared Atmospheric Sounding Interferometer (IASI) on board the MetOp-A satellite.



### 4.2.1 Comparison with NDACC ground-based FTIR measurements

A Bruker 125HR FTIR spectrometer started operating at the Maïdo Observatory in March 2013 with a primary dedication to NDACC measurements (Zhou et al., 2016). This NDACC ground-based FTIR observes the absorption of the direct solar radiation with high spectral resolution (0.0035-0.0110 cm$^{-1}$) and uses the pressure broadening effect of absorption lines to
retrieve volume mixing ratio (vmr) low vertical resolution profiles of target gases. The FTIR ozone measurements show a good sensitivity from the ground up to about 45 km. Within this vertical range, about 4 vertical layers can be distinguished, i.e. the vertical resolution varies from 8 to 15 km (Vigouroux et al., 2015). In this study, the FTIR retrievals are based on an optimal estimation method (Rodgers, 2000), carried out with the SFIT4 algorithm (https://wiki.ucar.edu/display/sfit4), which is an open source code, jointly developed at the NASA Langley Research Center, the National Center for Atmospheric Research
(NCAR), the National Institute of Water and Atmosphere Research (NIWA) and University of Bremen. HBr cell measurements are performed on a daily basis to verify the alignment of the instrument and to obtain the instrument line shape (ILS) using the LINEFIT14.5 program (Hase et al., 1999). The retrieval scheme is described in Vigouroux et al. (2015), and closely follows the recipe of the Jungfraujoch station (except for the ILS which is fixed from LINEFIT results at Maïdo): the retrieval microwindow is 1000-1005 cm$^{-1}$, the a priori data comes from the WACCMv6 model and pressure and temperature a priori
profiles were obtained from National Centers for Environmental Prediction. The a priori water profile is obtained from a dedicated pre-retrieval. Each ozone profile is retrieved with the signal to noise of the source spectrum. The total uncertainty on the ozone profile is dominated by the smoothing error (i.e. the poor vertical resolution of the profile), the temperature and the spectroscopic uncertainties. We use the following approach for comparison:

    i) FTIR performing observations during daytime, each LIO3T measurement is co-located to all FTIR measurements within
a 24-hour time window;

    ii) for each such a pair (114 pairs in total), the LIO3T profile is regridded (with mass conservation) to the FTIR height grid;

    iii) FTIR measurements are averaged within the 24-hour time window around a single LIO3T measurement for comparison;

    iv) at this stage we have a set a comparable pairs of measurements with various validity domain for LIO3T profiles; however, the method needs constant boundaries for the partial column used for comparison; we then choose the partial column shared
by a sufficient number of LIO3T profiles to allow a reasonable comparison; the upper and lower limits of this partial column are called hereafter "valid range for comparison";

    v) the regridded LIO3T profile is smoothed with the FTIR averaging kernel matrix and a priori (see, e.g., Rodgers and Connor, 2003; Vigouroux et al., 2008); to allow for the smoothing, the LIO3T measured profiles are extended by the FTIR a priori outside the valid range for comparison. By smoothing the LIO3T profiles, we degrade them to the FTIR low vertical
resolution, and we can get rid of the FTIR smoothing uncertainty in the uncertainty associated with the comparison;

    vi) finally, a partial column is calculated from this smoothed LIO3T profile in the valid range of comparison.

We find 11 comparison pairs over the studied period within the 8.5-16km valid range for comparison. In this 8.5-16km partial column, the ground-based NDACC FTIR has 1.1 degree of freedom (Rodgers, 2000) and a mean total uncertainty of





7.5%. Figure 10 shows the FTIR a priori profile and averaging kernels for this 8.5-16km partial column, both of them being used to smooth the LIO3T measurements to compare with the FTIR ones.

Figures 11 shows the comparison of the FTIR and LIO3T partial columns time series. One can see that there is a good agreement between the datasets considering the uncertainties. We find a $D$ of 11.8% between datasets (LIO3T high). Note that, due to the sparse comparison points, the southern hemisphere biomass burning season is not visible on this plot.

### 4.2.2 Comparison with IASI measurements

IASI is on board the MetOp-A satellite launched in a Sun-synchronous orbit around the Earth at the end of 2006. A second IASI was launched on board MetOp-B in September 2012 and the launch of the third one (MetOp-C) is planned for late 2018. In this comparison, IASI/MetOp-A data are used. IASI is a FTIR instrument that measures the thermal infrared radiation emitted by the Earth's surface and atmosphere in the 645-2760 $cm^{-1}$ spectral range with a spectral resolution of 0.5 $cm^{-1}$ apodized and a radiometric noise below 0.2K between 645 and 950 $cm^{-1}$ at 280K (Clerbaux et al., 2009).

IASI is an interesting instrument for our intercomparison effort as it provides global Earth coverage twice daily with overpass times at 09:30 and 21:30MLT (mean local time) and a nadir footprint on the ground of 12 km. IASI has significant sensitivity to tropospheric $O_3$. As LIO3T usually fires between 7PM and 1AM local times, we used here the IASI nighttime overpass measurements. The IASI data used in this study come from the FORLI-$O_3$ v20151001 scheme (Hurtmans et al., 2012; Boynard et al., 2016).

To compare measurements from both instruments, IASI retrievals are averaged over a 1°x1°box around the Maïdo Observatory location. We then use the same approach as described hereabove for the comparison with the ground-based FTIR observations (except points i) and iii)). We find 39 comparison pairs over the studied period within the 6-16km valid range for comparison. In this 6-16km partial column, IASI has 1.6 degree of freedom (Rodgers, 2000) and a mean total uncertainty equal to 18.4%. Figure 12 shows the mean IASI a priori profile and mean averaging kernels in the 6-16km partial column for the 39 comparison pairs. In the following, LIO3T measurements are smoothed according to these characteristics of the IASI retrievals.

Figures 13 shows the comparison of the IASI and LIO3T partial columns time series. We obtain a good agreement between the datasets considering the uncertainties. We find a $D$ of 11.3% between datasets (LIO3T high). These results are in agreement with the 5-15% $O_3$ abundance difference of IASI in the troposphere compared to ECC soundings reported recently by Boynard et al. (2016). Note that the biomass burning seasonal cycle is clearly visible on the time series of both instruments in this partial column.

## 5 Dataset and time series

The Maïdo Observatory lidars (LIO3T, LIO3S, LI1200 and Doppler wind lidars) are operated on a routine basis two nights a week and intensively several times a year in campaign mode. Five campaigns were carried out during this 3 year-period: MALICCA (MAïdo LIdar Calibration CAmpaign) 1 (April 2013) and MALICCA 2 (November 2013) (Keckhut et al., 2015),





LIDEOLE (LIDar EOLE) 1 (November 2014) and LIDEOLE 2 (September 2015) - dedicated to the Doppler wind lidar intercomparisons with wind soundings - and MORGANE in May-July 2015 (Portafaix et al., 2015; Vérèmes et al., 2015; Duflot et al., 2016a; Posny et al., 2016). Figure 14 shows the number of profiles (and hours of acquisition) per year; one can see that there are increasing numbers of profiles acquired each year, reaching a total of 84 LIO3T $O_3$ profiles for the entire

period. This increase is even more noticeable when looking at cumulative acquisition durations: from 28 hours in 2013 to 144 hours in 2015. This striking increase is explained by the system failures that occurred in 2013 and 2014 (KDP crystal and laser failures in September-November 2013, and Raman cell optics issues in May-July 2014), and by the huge effort made during the MORGANE campaign in 2015 to operate the LIO3T system. Indeed, one of the main goals of this last campaign was the validation of LIO3T $O_3$ against ECC soundings in view of the application of the lidar to a NDACC labellisation.

Figure 15 shows the cross histogram of the lower and upper limits of the LIO3T $O_3$ profiles validity domain. Lower limit ranges from 6 to 10 km amsl, and upper limit from 12 to 19 km amsl. One can see that most profiles start at 6 km and end at 17-18 km amsl.

Figure 16 shows the seasonal distribution of the measured ozone profiles. Note that nearly half of March-April-May (MAM) profiles were acquired during the MORGANE campaign (light blue bar in Figure 16). Note also the few number of profiles in

December-January-February (DJF) each year caused by the high cloudy skies occurrence frequency in the rainy season.

Figure 4 (bottom panel) shows the monthly averaged LIO3T $O_3$ profiles time serie. The same features as for the LIO3T$_{UR}$ and ECC climatologies can be noticed (cf. Section 2.2):

- ozone abundance increase during the southern hemisphere biomass burning season (June-December peaking in October at $\approx 9x10^{11}$ molec/cm$^3$ on average between 6 and 10 km amsl);

- ozonopause altitude decrease from $\approx$ 17km to $\approx$ 15km amsl in late austral winter-spring.

Looking at the three climatologies, an additional seasonal pattern clearly appears: the minimum of ozone abundance in February between 10 and 16 km amsl ($\approx 3x10^{11}$ molec/cm$^3$ on average), which is likely a signature of the austral summer deep convection bringing boundary layer-ozone poor air masses up to the mid-upper troposphere.

In conclusion, the three datasets show a remarkable - and reassuring - agreement in terms of patterns and values.

Figure 17 shows the seasonal profiles derived from the LIO3T $O_3$ measurements. The southern hemisphere biomass burning season is still clearly visible in the September-October-November profile (SON), with an increase that covers the whole probed column, and also on the June-July-August (JJA) profile from 6 to 13 km amsl.

# 6   Conclusions and future plans

A DIAL tropospheric ozone lidar was installed and operated on the Université de la Réunion campus site from 1998 to 2010.

It provided 427 ozone profiles over the 13 years of operation. In 2012, the system was moved up to the Maïdo Observatory and routine $O_3$ observations started in February 2013 by the LIO3T system. From then until January 2016, 84 $O_3$ profiles were acquired and LIO3T operation is ongoing. These $O_3$ measurements was recently affiliated in the NDACC family.



The LIO3T $O_3$ observation scheme is based on the DIAL technique, which currently detects two wavelengths, 289 and 316 nm, with multiple receivers. The transmitted wavelengths are generated by focusing the output of a quadrupled Nd:YAG laser beam (266 nm) into a pair of Raman cells, filled with high-pressure deuterium, using helium as buffer gas. With the knowledge of the ozone absorption coefficient at these two wavelengths, the range-resolved number density can be derived.

Optimal range for the actual system is 6-19km amsl, depending on the system performance and atmospheric conditions; for a 1-hour integration time, vertical resolution varies from 0.7 km at 6 km amsl to 1.3 km at 19 km amsl, and mean uncertainty over the 6-19km range is between 6 and 13%.

Comparisons with ancillary data were performed showing a good agreement between datasets considering the uncertainties: we found a 7.7% $D$ between LIO3T $O_3$ observations and 8 ECC sondes simultaneously launched from the Maïdo Observatory

(LIO3T low), 10.3% $D$ between LIO3T $O_3$ observations and 37 ECC sondes launched from the Gillot site during day time in a $\pm$ 24-hour window around lidar shooting (LIO3T low), 11.8% $D$ between LIO3T $O_3$ and 11 ground-based NDACC FTIR measurements acquired during day time in a $\pm$ 24-hour window around lidar shooting in the 8.5-16 km partial colum (LIO3T high), and 11.3% $D$ between LIO3T $O_3$ and 39 simultaneous nighttime IASI observations over Reunion Island in the 6-16 km partial column (LIO3T high).

ECC, LIO3T$_{UR}$ and LIO3T $O_3$ monthly climatologies all exhibit the same range of values and the same seasonal patterns:

- the ozone abundance increase in austral winter and spring due to the southern hemisphere biomass burning season;

- the ozonopause altitude decrease from late austral winter to early austral summer due to the winter time thermal effect on the troposphere thickness combined to the enhanced occurrence of STE in austral spring and summer;

- the ozone abundance minimum in late austral summer in the mid-upper troposphere due to deep convection uplifting

ozone-poor air masses from the boundary layer.

This tropospheric ozone data set covering the tropical free troposphere and UTLS of a sparsely documented region (South Western Indian Ocean) constitutes an extremely valuable resource for the validation of satellite tropospheric ozone retrievals, analysis of the ozone variability and sources, and for long term atmospheric monitoring.

Future plans for the LIO3T system are to (1) use the available 532nm residual beam to detect and study aerosols in the

free troposphere, TTL and lower stratosphere. The use of the infrared signal (1064nm) to study aerosols is also planned; (2) implement NDACC recommendations in the data processing (ozone cross sections, background and saturation corrections uncertainties propagation); (3) calculate uncertainties due to the presence of aerosols in the troposphere using an iterative aerosol assessment procedure, ideally using the 532nm backscattered signal.

*Acknowledgements.* The authors acknowledge the European Communities, the Région Réunion, CNRS, and Université de la Réunion for

their support and contribution in the construction phase of the research infrastructure OPAR (Observatoire de Physique de l'Atmosphère de La Réunion). OPAR is presently funded by CNRS (INSU) and Université de La Réunion, and managed by OSU-R (Observatoire des Sciences de l'Univers de La Réunion, UMS 3365). The authors also gratefully acknowledge E. Golubic, P. Hernandez and L. Mottet who are deeply involved in the routine lidar observations at the Maïdo facility. J. Witte (NASA/GSFC) is acknowledged for the ECC data reprocessing. IASI is a joint mission of EUMETSAT and the Centre National d'Etudes Spatiales (CNES, France). The IASI L1C data are distributed in near

real time by EUMETSAT through the EUMETCast system distribution. The authors acknowledge the Aeris data infrastructure for providing access to the IASI L1C data and L2 temperature data used in this study. This work was undertaken in the framework of the EUMETSAT $O_3$M-SAF project (http://o3msaf.fmi.fi), the European Space Agency $O_3$ Climate Change Initiative ($O_3$-CCI, www.esa-ozone-cci.org). The ULB French scientists are grateful to CNES and Centre National de la Recherche Scientifique (CNRS) for financial support. PFC is grateful

5   to Belspo and ESA (Prodex IASI.Flow project) for financial support. The colleagues from BIRA-IASB acknowledge the support from the Belgian Science Policy Office, as well as from ESA/PRODEX and the Copernicus programme (CAMS-VAL).





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



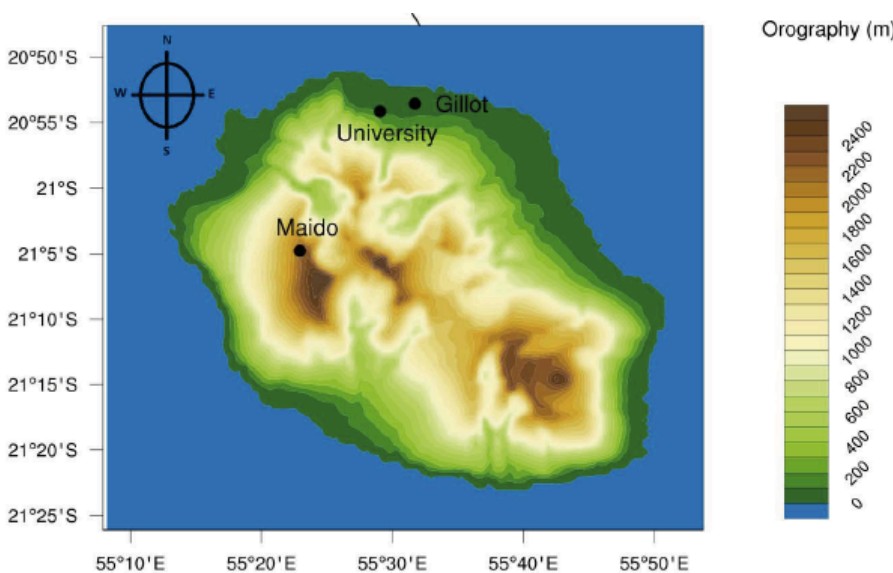

**Figure 1.** Map showing the locations of the different measurement sites, Maïdo Observatory, Gillot, and University in Reunion Island.

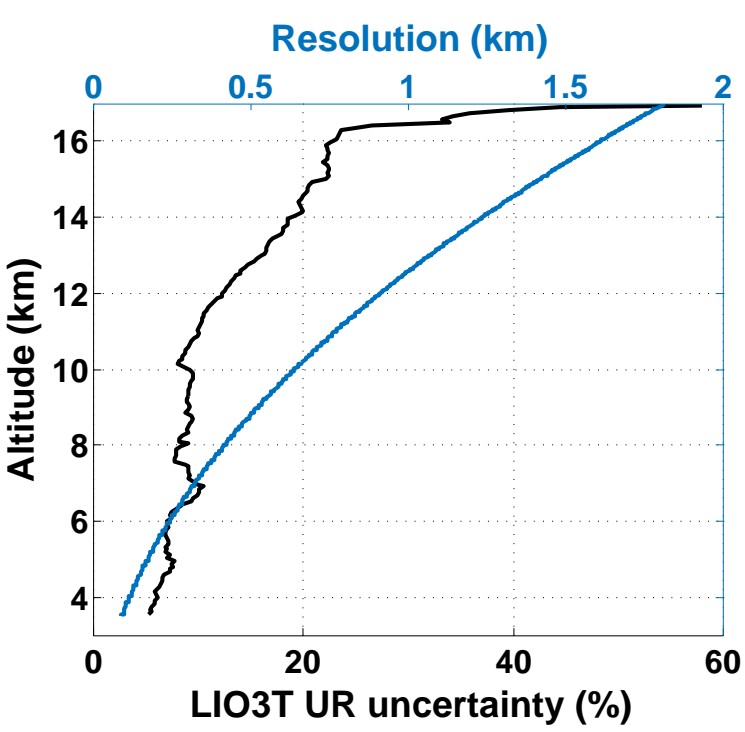

**Figure 2.** Mean LIO3T$_{UR}$ uncertainty (black curve) and resolution (blue curve) at Université de la R'eunion.





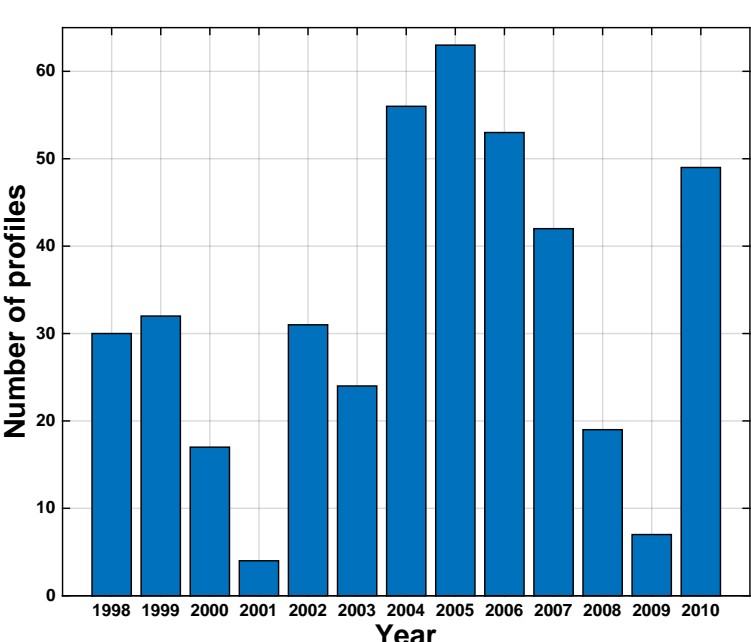

**Figure 3.** Number of LIO3T$_{UR}$ ozone profiles per year.



**Figure 4.** Top left panel: Monthly ECC climatology (2005-2015, Gillot site) between 0 and 19km amsl; Top right panel: Monthly LIO3T$_{UR}$ climatology (1998-2010, Université de la Réunion campus site); Bottom panel: Monthly LIO3T climatology (2013-2015, Maïdo Observatory).





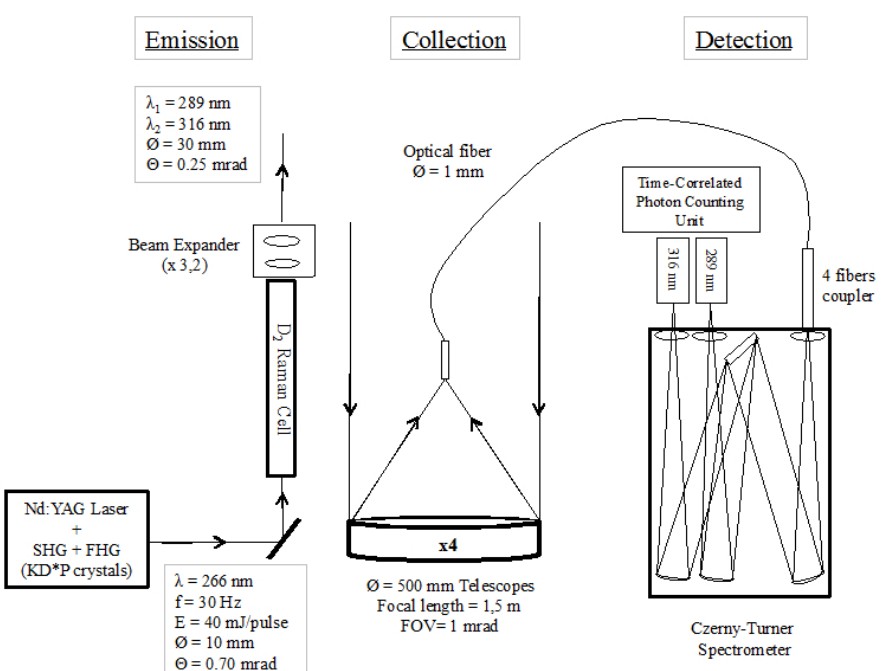

**Figure 5.** LIO3T O$_3$ DIAL Experimental schematic.





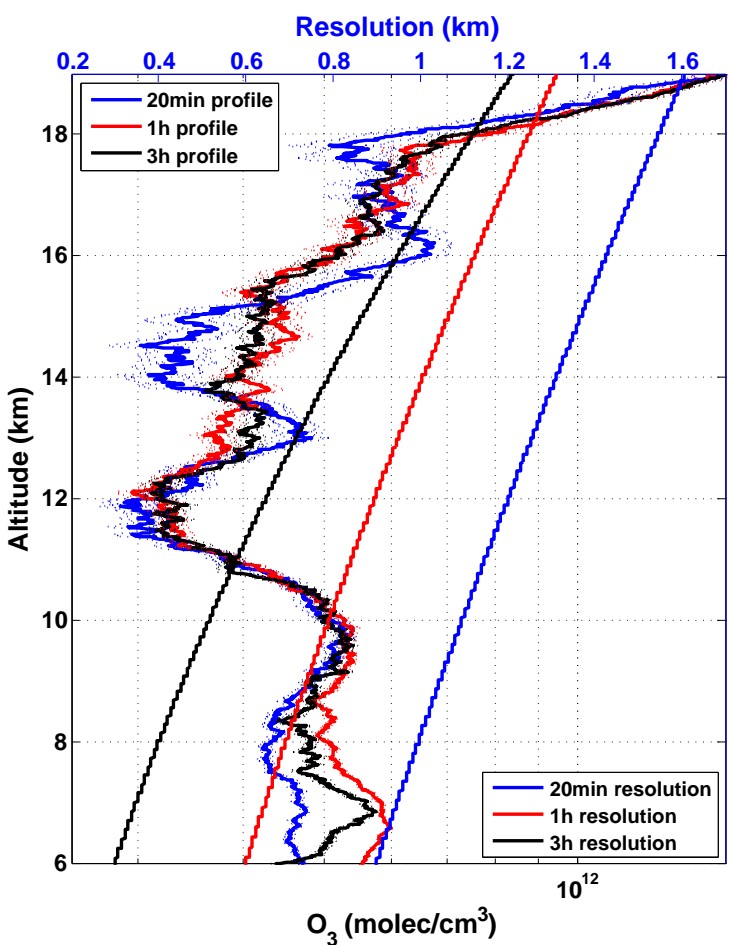

**Figure 6.** Lower X-axis: ozone profile obtained on the 2013/07/09 for a 20 minutes (blue curve), 1 hour (red curve) and 3 hours (black curve) integration time. Uncertainties for each profile are given by dotted curves. Upper X-axis: corresponding resolution profiles for a 20 minutes (blue line), 1 hour (red line) and 3 hours (black line) integration time.





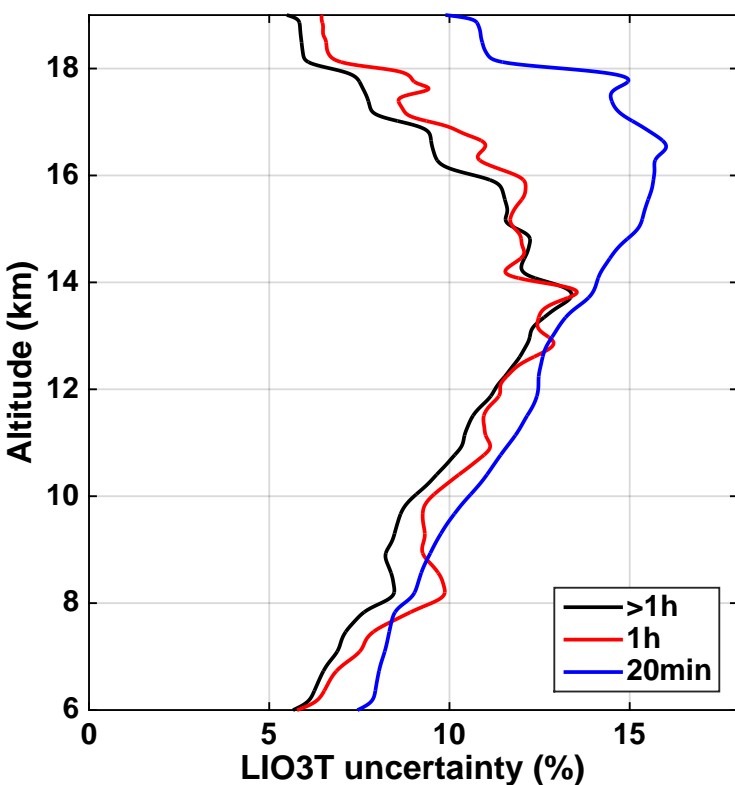

**Figure 7.** Mean uncertainties of the LIO3T ozone profiles for integration times greater than 1 hour (black curve), equal to 1 hour (red curve) and equal to 20 minutes (blue curve).




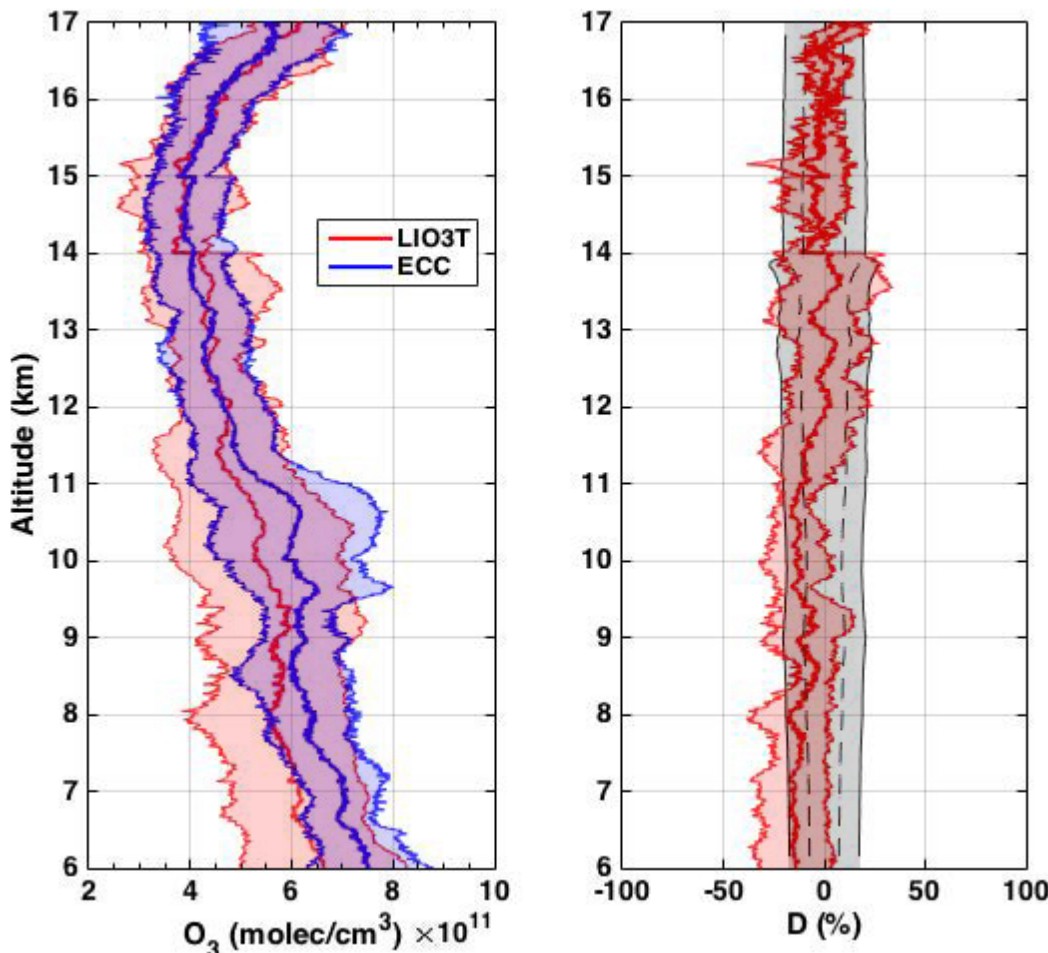

**Figure 8.** Left panel: mean LIO3T ozone profile (red curve) and mean ECC profile (blue curve) measured during the 8 intercomparison measurements performed at Maïdo. The shaded areas give the 1 standard deviation around the mean; Right panel: mean $D$ between the LIO3T and ECC mean profiles (red curve), mean LIO3T uncertainty around zero (black dashed curves) and mean LIO3T uncertainty + ECC accuracy around zero (black curves and black shaded area).





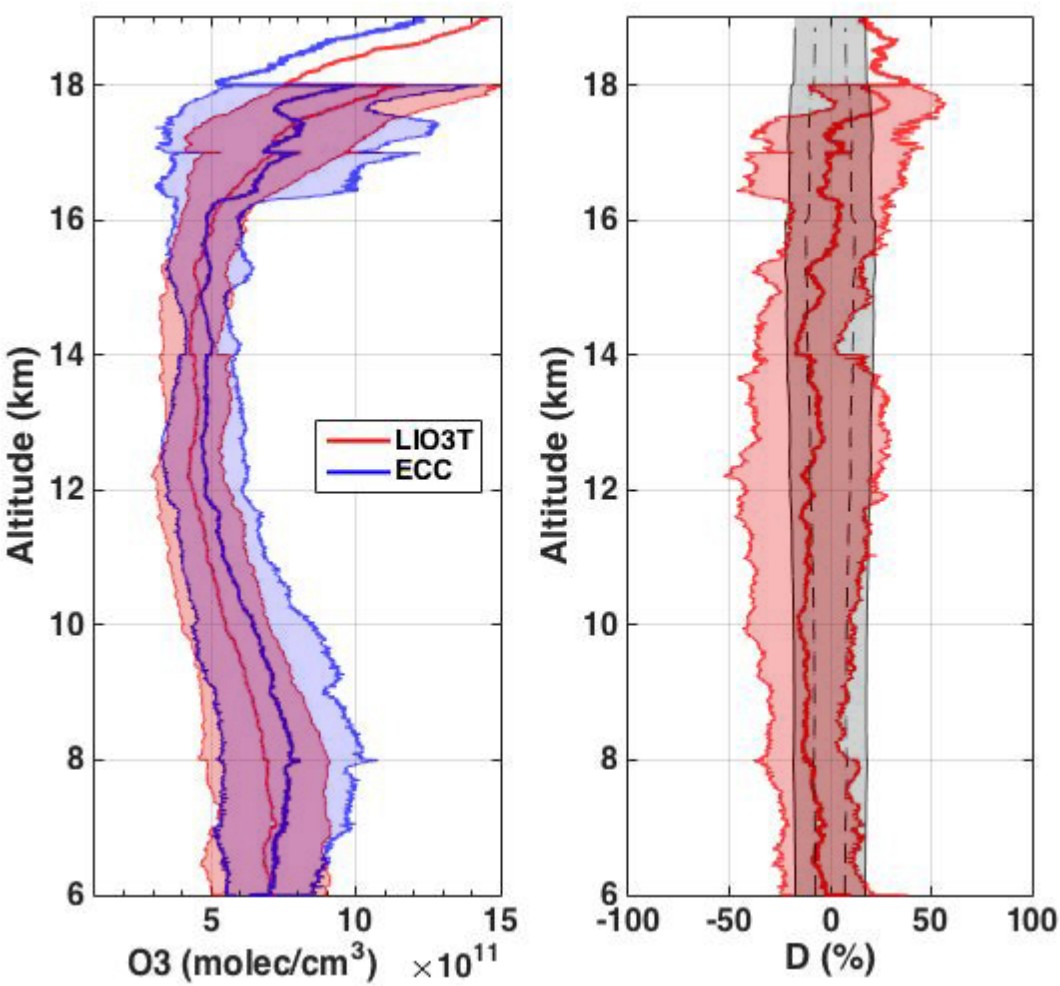

**Figure 9.** Same as figure 8 for NDACC/SHADOZ Gillot ECC soundings and "full night" LIO3T profiles.



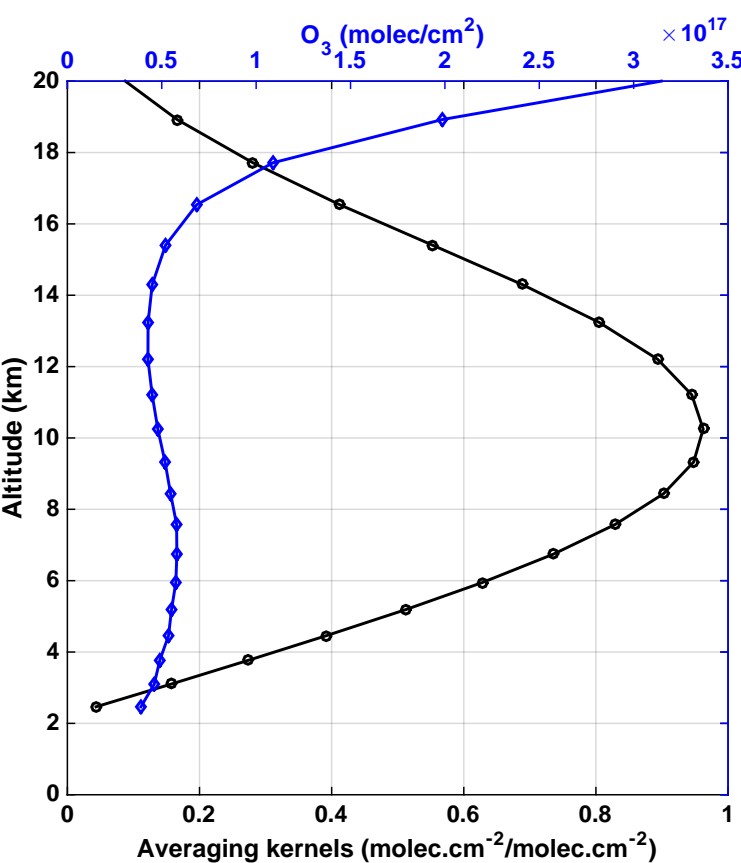

**Figure 10.** Lower X-axis: ground-based NDACC FTIR averaging kernels for the 8-16 km partial column (black curve and circles); Upper X-axis: ground-based NDACC FTIR ozone a priori profile (blue curve and diamonds).





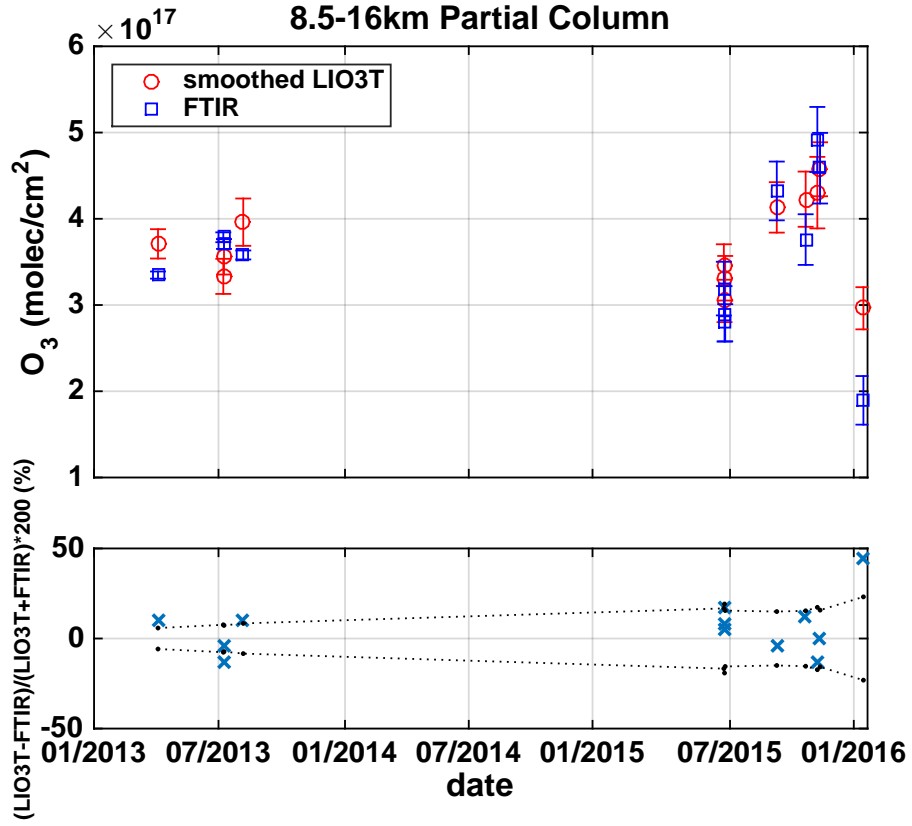

**Figure 11.** Upper panel: smoothed LIO3T (red circles) and ground-based NDACC FTIR (blue squares) 8.5-16km ozone partial columns. Vertical bars give uncertainties for each measurement; Lower panel: relative difference (%) between LIO3T and FTIR measurements (blue crosses) superimposed on LIO3T + FTIR uncertainties around zero (black dotted lines and dots).





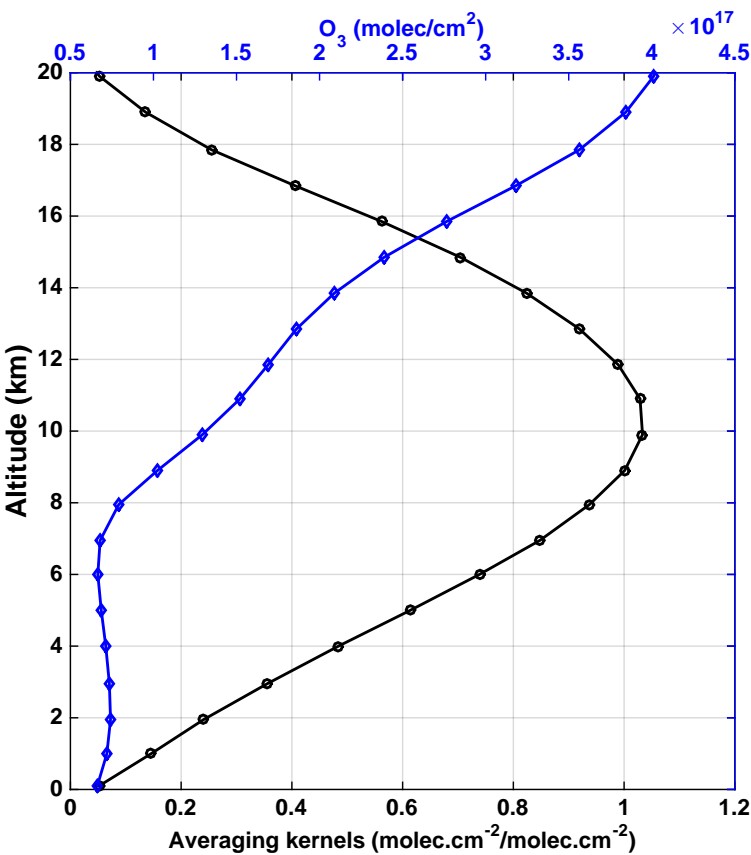

**Figure 12.** Lower X-axis: IASI averaging kernels for the 6-16 km partial column (black curve and circles); Upper X-axis: IASI ozone a priori profile (blue curve and diamonds).





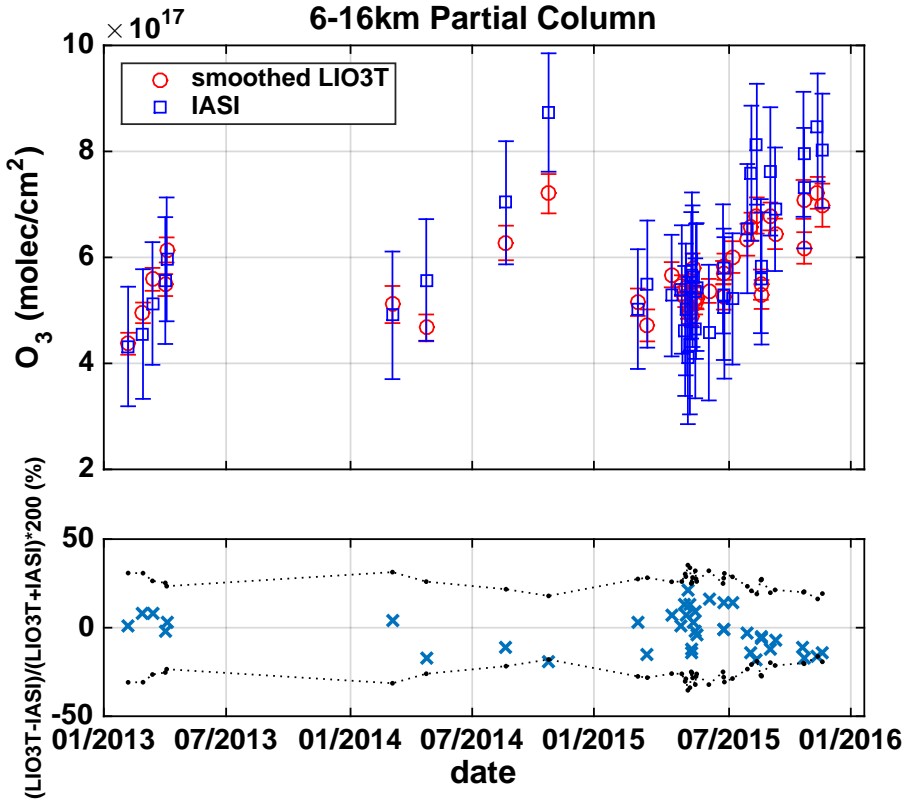

**Figure 13.** Upper panel: smoothed LIO3T (red circles) and IASI (blue squares) 6-16km ozone partial columns. Vertical bars give uncertainties for each measurement; Lower panel: relative difference (%) between LIO3T and IASI measurements (blue crosses) superimposed on LIO3T + IASI uncertainties around zero (black dotted lines and dots).





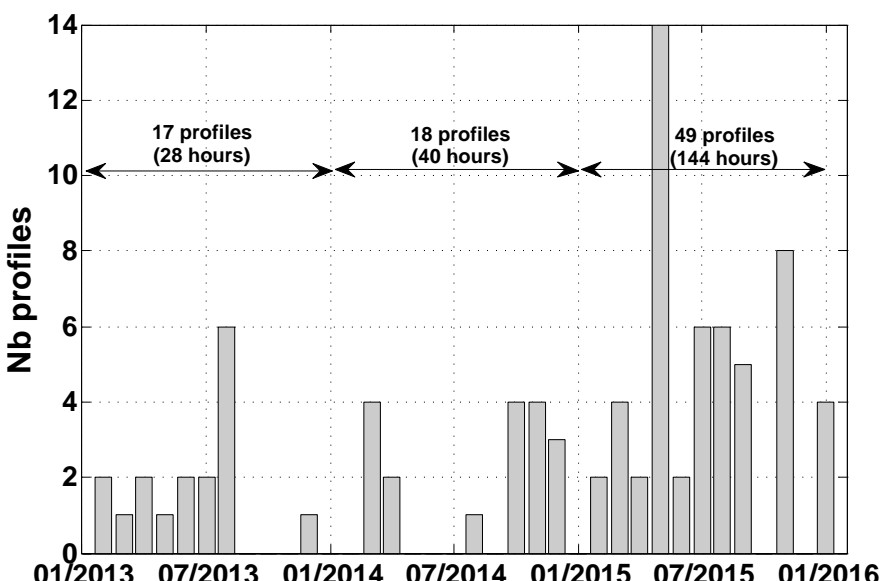

**Figure 14.** Monthly distribution of the LIO3T profiles from January 2013 to January 2016. Annual total numbers of profiles are given for each year together with cumulative acquisition durations.





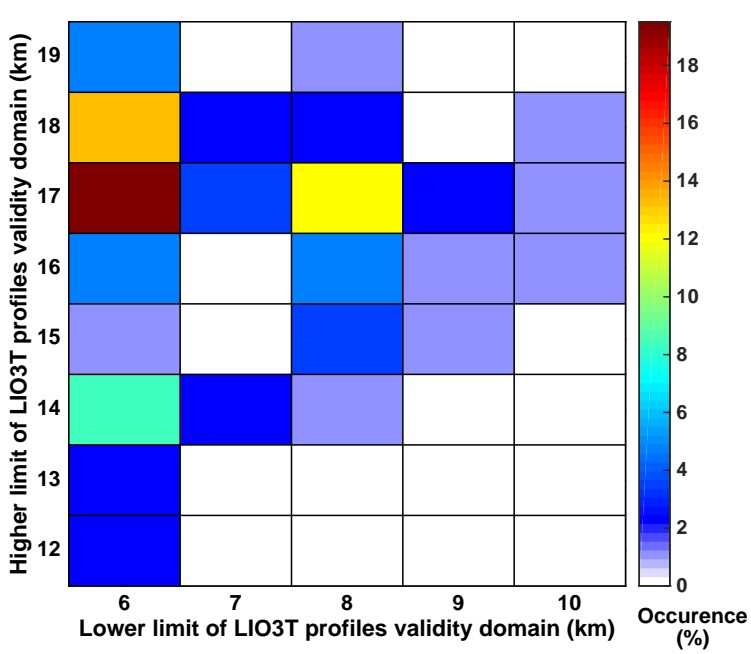

**Figure 15.** Cross histogram of the lower and upper limits of the LIO3T profiles validity domain (relative occurrence in %, 84 profiles).





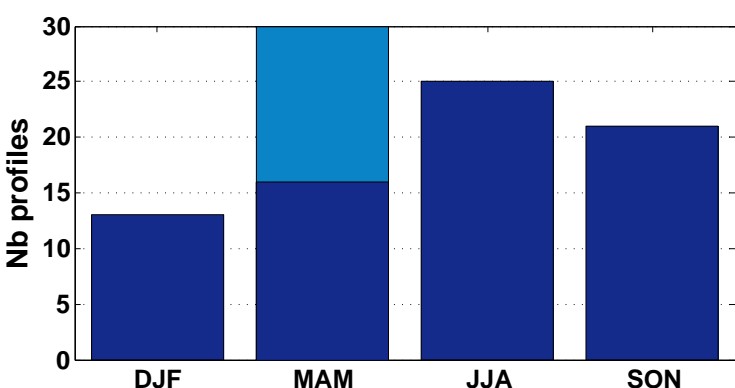

**Figure 16.** Seasonal distribution of the ozone profiles measured by LIO3T for December-January-February (DJF), March-April-May (MAM), June-July-August (JJA) and September-October-November (SON). The light blue bar in MAM gives the number of ozone profiles acquired in May 2015 during the MORGANE campaign.





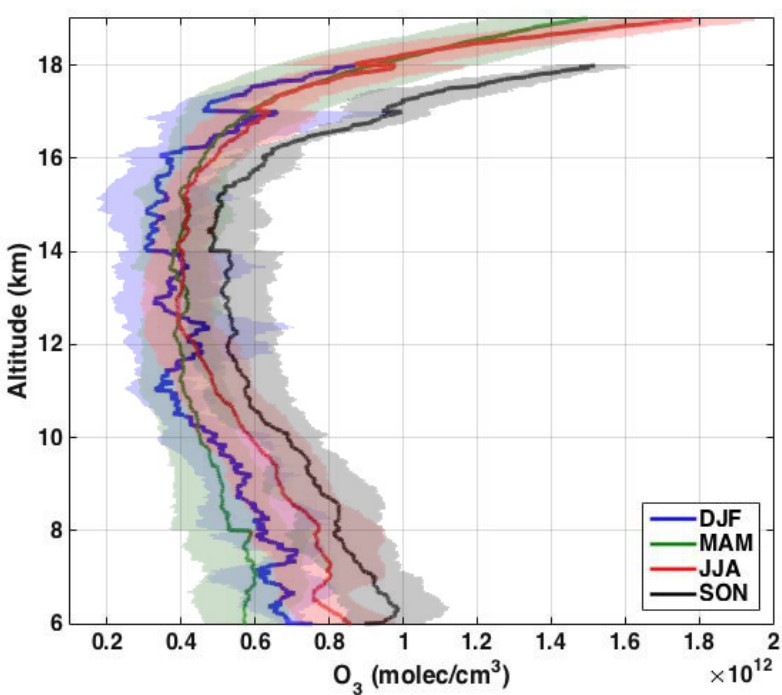

**Figure 17.** Seasonal LIO3T ozone profiles for DJF (blue curve), MAM (green curve), JJA (red curve) and SON (black curve). The shaded areas give the 1 standard deviation around the mean.



| | |
|---|---|
| Energy at 266 nm | 40 mJ/pulse |
| Laser frequency | 30 Hz |
| Beam diameter | 10 mm |
| Raman cells length and diameter in/out | 1500mm and 20/55 mm |
| Gases (pressure) in Raman cell | He (10 bar) and D (10 bar) |
| ON/OFF wavelengths | 289/316 nm |
| Emitted beam output diameter and divergence | 30mm and 0.25 mrad |
| Reception telescopes | 4 x 500 mm mosaic |

**Table 1.** Main LIO3T $O_3$ DIAL technical characteristics.





| Date | Profile valid range (km) |
|------|--------------------------|
| 2013/06/24 | 6-14 |
| 2013/06/25 | 6-14 |
| 2015/05/11 | 6-17 |
| 2015/05/15 | 10-16 |
| 2015/05/26 | 6-12 |
| 2015/05/28 | 6-17 |
| 2015/07/06 | 6-15 |
| 2015/07/07 | 6-17 |

**Table 2.** Dates of comparisons with collocated ECC soundings and corresponding LIO3T $O_3$ profile valid ranges.