# Peer review of "Ozone profiles by DIAL at Maïdo Observatory (Reunion Island) Part 1. Tropospheric ozone lidar: system description, instrumental performance, and result comparison with ozone external data set"

_Atmospheric Measurement Techniques, 2016_

## Referee Comment (RC1) · Anonymous Referee #2 · 31 Jan 2017

Formal Review for:

Ozone profiles by DIAL at Maïdo Observatory (Reunion Island) Part 1. Tropospheric ozone lidar: system description, performances evaluation and comparison with ancillary data.

Scientific Summary: It's critically clear that the Maïdo Observatory is uniquely located to investigate constituent (e.g. ozone) variability at various levels within the atmosphere. A new/improved tropospheric ozone lidar is described at this site, in conjunction with a long-term data set of the previous version of the tropospheric ozone lidar. The manuscript is well written and logical and I recommend it for publication after some minor revisions suggested below.

General Comments:

It should be reiterated throughout the text which lidar system (old vs. new) is being described. Although the subscript "_UR" is used to denote the old system, I still found several instances confusing as to which lidar data was presented.

Figure 1: Recommend adding ECC, LIO3T, and LIO3t_UR to the map after site location. Also altitude site altitude would be helpful.

Figure 2: The VR and uncertainty are effectively presented in Figure 6/7 for the new instrument. The information in Figure 2 could be simply stated in the text at various altitudes or added as a component to complement figure 6/7. Or conversely, bring in the new lidar VR/Uncertainty budget here as well to simultaneously show the absolute uncertainty as well as improvement to the system.

Figure 3: It would be useful to add the occurrences (with a different color) for the new system. It is stated later in the text and elaborated in Figure 14, but may make more sense to have towards the beginning to complement the old lidar system. Conversely, you could move all of the lidar/ECC discussion to the end (where Figure 14 currently exists).

Section 2.2 and Figure 4: Does the missing data (e.g. December above 10 km) in bottom panel Figure 4 correspond to lack of lidar coverage? Or the uncertainty of the measurement was larger than a certain threshold? It would be helpful to recreate the occurrences in Figure 3 as the same layout as Figure 4 but monthly instead of yearly. Generally the ECC shows higher ozone between 10-16 kms, regardless of time of year. Is this a known issue? Relevant to sonde reprocessing? and more importantly, are these differences within the uncertainty of the lidar measurement (with a set vertical resolution/temporal averaging)? A difference plot may highlight potential differences. Is the ECC climatology only from Gillot or combination of Gillot/Maido?

Figure 6: Are the differences observed between 6-8 km geophysical or based on the algorithm/processing? Is there a sounding for this observational period to compare with? I would assume the SNR to be largest from 6-8 kms and therefore the temporal averaging would generate less obvious differences. Is it possible there is a partial overlap or saturation correction that needs to be addressed?

L25 – Remove "5The"

Figure 7: Are these uncertainties explicitly based on the retrievals performed in Figure 7 or are these average uncertainties over many nights. It seems odd that at 8 kms, there is a greater uncertainty from a 1 hr measurement than at 20 mins. Please clarify.

P8L23 – Is Witte et al., still in prep or under review?

P9L9 – Should be McGee, not MacGee

Figure 8/9: There appears to be the largest difference between the lidar and ECC below 8 kms. Is this systematic for all 8 launches? Is this also apparent in the original lidar data? Were these sondes launched from Gillot or Maido? It is useful to have a profile of the number of comparisons to make discontinuities apparent. If the potentially contaminated (e.g from aerosols) data were removed would the results improve?

Eq 6, what is reference for the 200 in the numerator? Is this based on sample size?

Figure 11: What's driving the difference in FTIR uncertainty from the beginning of the time series to the end? Is this the full uncertainty or partial uncertainty? Bottom panel y-label can just be D.

Further suggestion #1: Instead of the cross histogram in figure 15, show a complete time series of all the data in the archive used to make this plot. Showing the reader the entire data set instead of the histogram will emphasize the observed variability in O3.

Further suggestion #2: One very unique piece of information at the lidar site location is the ability to observe this enhanced O3 related to biomass burning. In the section describing the IASA comparisons, it may be helpful to show a world map of one day (or gridded month) where Maido is directly in the center of a CO transport plume, therefore highlighting the location and detection capabilities of the instruments.

---

## Referee Comment (RC2) · Anonymous Referee #3 · 22 Feb 2017

This paper aims to be a reference for the further use of data acquired by a DIAL tropospheric ozone lidar recently installed at Maïdo Observatory (Reunion Island). Authors describe first the lidar historical context and technical evolutions. Secondly they compare the ozone profiles data set with ECC sondes, with data from a ground based FTIR spectrometer and with the space-borne IASI data. They provide (monthly averaged and seasonal) climatology and time-series to describe the ozone concentrations from the surface up to 19km (maximum) and conclude on the agreement.

General comments :

This paper is indeed well in the scope of the journal, but needs to be improved before acceptance. The manuscript appears as not enough mature, missing frequently of accuracy with lack of definitions and arguments. Provide sections 3.2.1, 3.2.2 and 3.2.3 earlier in the manuscript because these informations should be available for the lidar when operated at l'Université de la Réunion. Equation 6, correlated text and figures are unclear and I will not address any comments there, unfortunately a key point in your study. Improve Figures and to suppress the figures or table when not enough informative and avoid repetitions. Material should be valuable with a real effort and some work to reinforce the demonstration and the benefits/limits from this new instrumentation.

Specific comments (not extensive) :

Title should be revised discarding Part 1 which is not meaningful. I suggest : "Ozone profiles from a DIAL lidar at Maïdo Observatory (Reunion Island): instrumental description, instrumental performance, and result comparison with ozone external data set.

Your paper is based on a DIAL instrument. . . but DIAL is nowhere defined as a Differencial Absorption Lidar. Please define this acronym at least once.

This lidar technique is now well known and you refer to major technical points already published regarding your instrument (ex line 31 p3-line 1 p4) and data processing. Highlight what is new regarding LIO3T instrument and performance as compared to previous published papers. Please discuss the uncertainty from your figure 2 versus figure7 and explain why the results are similar/different with respect to altitude.

Homogenize your instrument labelling. Fix it once and make it consistent all along the text, including abstract, figures and tables. Along the text, you used : ozone lidar system (LIO3TUR), LIO3T, LIO3T O3, LIO3TUR, system, current system, current LIO3T system, LIO3T system, LIO3T lidar, LIO3T low and LIO3T high (this two latest are undefined elsewhere and used also in the abstract - unclear). For example I suggest to

**AMTD**

replace sentence starting line 10 page 3 by : "In the following, the lidar installed at the Université de la Réunion will be referred as LIO3TUR whereas the one installed at the Maïdo Observatory will be referred as LIO3T". Two labelling seems enough. If you need more, clearly explain.

Revise and Clarify line 5 p2 : "Ozone is a major greenhouse gas in the upper tro-posphere and lower stratosphere". Insist on radiative forcing contribution and remove "major"...

Rephrase lines 14-15 p2 and justify "great interest" to document climate change.

Line 20 p2 : "To improve the operation of remote sensing instrument" remote sensing is imprecise. Do you refer to space-borne instrumentation? ... and operation seems not appropriated. Replace "a 2160m-high atmospheric facility" by "a high atmospheric facility" to avoid repetition. You provide afterwards geographical coordinates and alti-tude.

Figure 1 : Gillot in black is difficult to see and altitude would be here welcome (figure already published please provide reference). Additional general comment on altitude: mention once all altitudes in your manuscript will be provided in amsl and avoid to repeat latter.

Replace title section 2 by "Historical context of the lidar installed at the Université de la Réunion (LIO3TUR, 1998-2010)

Replace title section 2.1 by "Instrumental description" or "Instrumental characteristics"

Line 7 p4 : First sentence is really too short... Is that your result? Could you explain the altitude range limitation? Don't forget your goal Line24 p3 is to provide data "over the entire tropospheric column...".

Figure 2 : by "resolution" you mean "vertical resolution"... What is the criteria for uncer-tainty? It is defined lately for LIO3T. It would be valuable in addition to have uncertainty expressed with the ozone unit provided. Explain why such a change in the uncertainty

[Figure]

of LIO3TUR with respect to altitude. Please modify your X-axis (top and bottom) giving a more precise scaling (more minor ticks).

Line 9 p4 : "Temporal resolution ... was chosen depending on atmospheric conditions"... Please justify what is your criteria.

Figure 3 and Figure 4 : please provide consistent figures. The number of profiles given per year is non adapted and consistent to support a monthly-averaged climatology. Note than < 10 profile per year is extremely low. Provide the monthly statistics for the ECC sondes, the LIO3TUR and LIO3T in a panel of your figure 4 , deep gaps within years could be specified in the text and you might suppress figure 3. Please replace figure 4 caption by "Monthly O3 climatology derived from ECC sondes over 2005-2015 at Gillot site between 0 and 19km (Top left panel), from LIO3TUR over 1998-2010 at Université de la Réunion campus site between X and XXkm (Top right panel) and from LIO3T over 2013-2015 at Maïdo Observatory between X and XXkm (Bottom panel).". Comparing the ECC sondes and LIO3TUR climatology, your figure 4 points out within 10-16km greater ozone concentrations from the ECC sondes... Explain, please.

Replace title section 3 by "The lidar installed at the Maïdo Observatory (LIO3T, 2013-present)"

Line 31 p4 : "Many instrument" is very imprecise.

Line 32 p4 : "another lidar" is imprecise. Which one : the LI1200 ?

Replace 3.1 in title "current system". In this paragraph, a similar description has been provided in Baray et al. (2013) and citation is missing. Is there something new? Suppress lines 18-20 from this paragraph, probably the right place would be in "conclusions and future plans". Please consider Table 1 and Figure 5 and try to avoid repetition between both. Table 1 should be probably suppressed. Please modify in Figure 5 caption using "LIO3T instrumental schema" ... I expect you are referring here precisely to LIO3T at Maïdo Observatory???

Line 23 p5 : "two backscattered lidar signal at two different wavelength" Check if correct.

Line 24 p5 : be more concise. I suggest "...wavelength, at 289nm (lon) where ozone is strongly absorbed and at 316nm (loff) where ozone absorption is weaker".

Line 28 p5 to Line 7 p6 : please avoid repetitions. Fix once z as the altitude and l as the wavelength. Please provide the interfering gas.

Line 7 p6 : This last sentence introduces further 3.2.2, specify. Additionally replace "this" by the equation number to be precise.-

Replace 3.2.2 title I suggest to replace by Saturation, correction and consequences on the vertical resolution

Line 9 p6 : "Saturation" what is saturated and what is the cause and why below 7km.

Line 13 p6 : 'etc'. Rephrase the sentence"... type of filter such as ....

Line 14 p6 : replace "described" by extensively detailed.

Line 15-16 p6 : "we decide"... replace by we found and rephrase but add (not shown) ...

Line 18 p6 : "the resolution rises" Modify expression. Compare the vertical resolution to figure 2 and explain.

Figure 6 : Modify the resolution lines using dashed lines and modify caption on the figure in order to discriminate ozone profiles and the vertical resolution given with respect to integration time. In the figure caption, origin of data (LIO3T) is missing.

Line 26 p6 : Please check here "to 5The"

Line 9-10 p7 : "new tropospheric ozone version..." the details on cross-sections and uncertainty you provide is not much informative and information seems different from line 25-26 p6.

Line 11-17 p7 : not at the right place, better place could be in conclusions and future plans

Line 25-26 p7 : explain why only at night. Explain why such integration time changes with respect to the data you compare to.

Line 28-30 p7. Please rephrase this sentence. "Notably" is imprecise, how much and where, but note the consistency within ∼9-11,5km and explain.

Line 4 p8 : "goes" replace by varies

Figure 7 : Why LIO3TUR uncertainty increases by a factor of 3 at least above 13km whilst the one of LIO3T decreases above 14km whatever the integration time by a factor of 2 at least.

Revise section 4 title. Comparison is really too short here. What pieces are you comparing.

Line 9 p8 'validate' Do you compare, evaluate or validate?. . . Keep constant and take care when using validation concept. I would say here you evaluate. . .

Line 13p 8 add IASI after space-borne and replace instrument by data.

Line 14-16: This paragraph is absolutely obscure. . . I don't understand at all what you are dealing with and description of the equation terms is hard to follow. Be clear, concise, avoid repetition and be simple, that will help. I can't go further here. . . Explain clearly how can D be negative and give a clear definition for D ??? What is LIO3Tn? What is MCDn? What is D by the end?

Line 17 p8 add sondes after ECC.

Line 23-28 p8 : For that reason I think you should use only 'evaluate' and not 'validate'. . . as mentioned above.

Line 9 p9 "SO2 loading too strong" imprecise, please provide informations on the

amount.

Line 11 p9 : "mean D" see above comment and remove LIO3Tlow which is undefined and that I do not understand. Clear definition is mandatory.

Line 15 p9: "Enhanced" by how much and provide a reference.

Line18-19 p9: Not at the right place, move to conclusions and future plans.

Figure 8 and 9 : I am not able to make any comment at the moment. Please use different line thickness for the mean and standard deviation in the figures. Text of the figure caption might be improved. Use same Yaxis altitude range in both figures.

Line 21-22 p9 : Replace 3PM by 15:00:00 LT, 7PM by 19:00:00 and 1AM by 01:00:00 LT

Line 25 p9 : replace "goes"

Line 29-30 p9: rephrase this sentence, unclear.

Line 2 p10 : This FTIR spectrometer should be added to the list of instruments operating at Maïdo Observatory and provided in introduction (line 26-27).

Line 19-20 p10 : improve text. . . both instrument are operating at the same place so their measurements are colocated. . . What has to be pointed out is the time window you consider for the comparison. Add "Thus, for the LIO3T comparison with FTIR, 114 LIO3T profiles are available".

Line 21 p10 : just mention the selected LIO3T profiles are regridded consistently to the FTIR.

Line 22 p10 How many FTIR data are averaged within 24h.

Line 23-31 p10: Very very hard to follow...

Line 3 p11: replace "time series" by "available over the 01/2013-01/2015 period."

[Figure]

Figure 11 : The caption is imprecise and bottom plot Y-axis text is not consistent with equation 6. Is it D??? Moreover, I found 12 symbols on this bottom plot ???

Figure 12 : I wonder if figure 10 and 12 should be gathered... Provide minor ticks for the month on the Xaxis plots

Line 18 p11 : suppress " hereabove for the comparison with the ground-based FTIR observations" and replace by "in section 4.2.1".

Figure 13: refer to figure 11 comment

Line 27 p11 : I don't clearly see a seasonal cycle, I see an O3 increase particularly in 2013 and 2016 suggesting an impact of biomass burning...

Section 5 : Title is data set and times series ... You are dealing here with data from Intensive period of observations during campaigns... I think it makes the difference with material in previous sections, if I well understood. Please provide a more explicit title. This section is lengthy.

Line 30 p11 – line 3 p12 : You provide LIO3T results in this section... Thus other lidar and details not used further are out of the scope of your study. Thus suppress. This text is very long and the Figure 14 do not provide more informations with respect to the text and is difficult to read. Shorten and improve text.

Figure 15 is not much informative and text seems to repeat what is given in Line19-21 p7.

Figure 4 : Please specify in the caption of bottom panel what the LIO3T climatology includes (data routinely performed and from intensive period of observations ???).

Figure 16 : please provide these informations (4 numbers) on Figure 17 and suppress Figure 16.

Line 16-20 p12 : bring in the light what are the benefits from your new lidar... For sure a better description from the upper-troposphere/lower-stratosphere than when located

at UR.

Line 21-24 p12 : valuable comments and for such reasons I encourage authors to carefully and rigorously revised the manuscript. Take care to the ECC caveats already mentioned.

Figure 17 : I recommend to add informations on the seasonal sampling frequency with respect to altitude and this should be done here with an additional panel.

Section 6 : bring more in the light the benefits provided from LIO3T... if monthly climatology from ECC is equivalent and to LIO3TUR and LIO3T (i.e. range of values and seasonal patterns). Could you reinforce you study here... Your goal was to describe the whole tropospheric column with LIO3T... What is your conclusion?

Line 15-20 p12 : specify altitude range here.

Line 29-32 p12 : suggestion : A DIAL tropospheric ozone lidar was operating on the Université de la Réunion campus site from 1998 to 2010, providing 427 ozone profiles. Note that this information on 427 profiles was not mentioned before. Same remark for LIO3T profiles. Replace "familly" by network.

Line 9 p13 : "we found a 7.7% D between"... revised with the D definition.

---

## Referee Comment (RC3) · Anonymous Referee #1 · 22 Mar 2017

The paper describes one of the instruments now making regular measurements of tropospheric ozone at a high-altitude, sub-tropical Southern Hemisphere observatory on Reunion Island in the Southern Indian Ocean. The location of the observatory is in a region of the globe that is under-represented in regular atmospheric observations. The paper is written to demonstrate the quality of the measurements and, as such, there importance in adding to the understanding of global ozone concentrations. The paper provides a chronology of measurements on the island and describes the advantages of the mountain site over the previous sea level site. It is well written and deserving of publication.

[Figure]

Page 2. Several sites are discussed here, Gillot, near the coast, where sondes are launched; the Universite' de la Reunion, where a lidar was first installed, and the Maido site. It would be helpful to have a small Table here with the Lat., Lon., Altitude and distances between the sites. It puts the information in a single location. In the related Figure 1, the color scheme makes some of the print difficult to read – this may be exacerbated in the printing process.

Pages 3 and 5. In the discussion of the Raman cell it is not clear if the laser beam is focused into the cell or not. My reading of this (and Figure 5) would lead me to believe that it is not, but this should be stated if it is the case. Also some information as to why such a choice was made. What is the efficiency of the conversion into each of the Stokes lines? D2 pressure in the cell? Here too, a small table with the input and output parameters of the Raman cells for each of the two systems is helpful in understanding any differences noted in the measurements.

Page 4, Line 27: "decrease" should be "decreases"

Page 5, Line 14: It states that a Hamamatsu 9980-110 PMT is used. I could not find a datasheet for a -110 variant of the R9980. Should this be an R9880-110? The quantum efficiency of the R9980 is quite low below 300 nm, whereas the R9880 is significantly higher. If this is a 9880-110 is it used at the 289 wavelength? This tube is susceptible to signal induced noise in the UV. The R7400 has much better characteristics than the 9880-110 at these wavelengths, but it has less gain.

Page 6, Line 26: should this end "with an uncertainty of 5%."?

Page 7, Line 19: Mount should be Observatory

Page 7, Line 23: Delete "Laser and Raman Cell", insert "at the transmitted wave-lengths,"

Figures 6,7 – Is the fact that there are increasing vertical resolutions as the integration time increases (Figure 6), responsible for the higher uncertainty for the one hour

integration, compared with the 20 minute integration? How are the vertical resolutions determined? Is a maximum desired uncertainty used to select the vertical resolution?

Figure 8: There is a discontinuity seen at 14 km in both panels. Is this also due to some measurements not reaching beyond this altitude? Was this sonde or lidar related? Figure 9 shows remnants of the same discontinuity.

Page 8, Lines 24 – 28: Was there a reason for using the non-standard solutions in the sondes at Gillot? If so this should be stated.

Page 9, Line 5: Insert "The" before "valid range".

Page 9, Line 7: delete "until", insert "near"

Page 9, 14-19. If a Morgane campaign curtain plot of the lidar retrieved aerosol scattering ratio is available, this would be a good place to insert that to go along with the discussion of the volcanic plume. At what altitude were the stratospheric intrusions located. Is this enhanced aerosol visible in the daily ASR plots? This should show up in the figure mentioned here.

Page 10, Line 19: "co-located" should be changed to "compared"

Page 10, Line 23: "set a comparable" should be "set of comparable"

Page 12, Line 9: delete "to a NDACC labellisation" insert "for inclusion within NDACC"

---

## Author Comment (AC1) · 20 Apr 2017

Dear Reviewer,

We would like to thank you for the insightful and helpful comments and suggestions. Please find in the supplement file the point by point response as well as the new version of the manuscript.

Best regards, Valentin Duflot et al.

Please also note the supplement to this comment:

[Figure]

http://www.atmos-meas-tech-discuss.net/amt-2016-403/amt-2016-403-AC1-supplement.zip

---

## Author Comment (AC2) · 20 Apr 2017

Dear Reviewer,

We would like to thank you for the insightful and helpful comments and suggestions. Please find in the supplement file the point by point response as well as the new version of the manuscript.

Best regards, Valentin Duflot et al.

Please also note the supplement to this comment:

http://www.atmos-meas-tech-discuss.net/amt-2016-403/amt-2016-403-AC2-supplement.zip

---

## Author Response (AR1)

*We would like to thank the reviewers for the insightful and helpful comments and suggestions. The response to each comment is below in italics.*

**RESPONSE TO REVIEWER 1**

The paper describes one of the instruments now making regular measurements of tropospheric ozone at a high-altitude, sub-tropical Southern Hemisphere observatory on Reunion Island in the Southern Indian Ocean. The location of the observatory is in a region of the globe that is under-represented in regular atmospheric observations. The paper is written to demonstrate the quality of the measurements and, as such, there importance in adding to the understanding of global ozone concentrations. The paper provides a chronology of measurements on the island and describes the advantages of the mountain site over the previous sea level site. It is well written and deserving of publication.

Page 2. Several sites are discussed here, Gillot, near the coast, where sondes are launched; the Université de la Reunion, where a lidar was first installed, and the Maido site. It would be helpful to have a small Table here with the Lat., Lon., Altitude and distances between the sites. It puts the information in a single location. In the related Figure 1, the color scheme makes some of the print difficult to read – this may be exacerbated in the printing process.

*Figure 1 is changed and Table 1 gives the coordinates and altitudes of the measurements sites as well as their distance from the Maïdo Observatory.*

Pages 3 and 5. In the discussion of the Raman cell it is not clear if the laser beam is focused into the cell or not. My reading of this (and Figure 5) would lead me to believe that it is not, but this should be stated if it is the case. Also some information as to why such a choice was made. What is the efficiency of the conversion into each of the Stokes lines? D2 pressure in the cell? Here too, a small table with the input and output parameters of the Raman cells for each of the two systems is helpful in understanding any differences noted in the measurements.

*The LIO3T and LIO3TUR rely on the same design. The Raman cell is equipped with two silica window lenses that had a focal distance of 75 cm to focus the beam as closely as possible to the center of the cell for optimal stimulated Raman scattering. Details regarding this Raman cell as well as the efficiency of the conversion into each of the Stokes lines are stated in Baray et al. (1999). We do not state this in the article as it is detailed in previous published papers.*

Page 4, Line 27: "decrease" should be "decreases"

*Done.*

Page 5, Line 14: It states that a Hamamatsu 9980-110 PMT is used. I could not find a datasheet for a -110 variant of the R9980. Should this be an R9880-110? The quantum efficiency of the R9980 is quite low below 300 nm, whereas the R9880 is significantly higher. If this is a 9880-110 is it used at

the 289 wavelength? This tube is susceptible to signal induced noise in the UV. The R7400 has much better characteristics than the 9880-110 at these wavelengths, but it has less gain.

*This is correct: this is a R9880-110 (not a R9980-110) and it is used for the 289nm channel. It is corrected page 6 line 7. R7400 could indeed be used with benefits for the 289nm detection channel, and, following your comments, this is something we plan to do in a near future.*

Page 6, Line 26: should this end "with an uncertainty of 5%."?

*Corrected.*

Page 7, Line 19: Mount should be Observatory

*Corrected.*

Page 7, Line 23: Delete "Laser and Raman Cell", insert "at the transmitted wavelengths,"

*Corrected.*

Figures 6,7 – Is the fact that there are increasing vertical resolutions as the integration time increases (Figure 6), responsible for the higher uncertainty for the one hour integration, compared with the 20 minute integration?

*Uncertainty increases when integration time decreases because of the increasing detection noise (i.e. decreasing signal to noise ratio).*

How are the vertical resolutions determined? Is a maximum desired uncertainty used to select the vertical resolution?

*Vertical resolution is not selected through a uncertainty threshold, but is calculated as described in Section 2.2.*

Figure 8: There is a discontinuity seen at 14 km in both panels. Is this also due to some measurements not reaching beyond this altitude? Was this sonde or lidar related? Figure 9 shows remnants of the same discontinuity.

*The "discontinuities" in the mean profiles shown on Figure 8 and 9 are caused by the varying valid ranges in the LIO3T profiles. It is stated page 8 lines 7 and 33. Moreover, Figures 5 and 6 now show the number of LIO3T profiles with respect to the altitude.*

Page 8, Lines 24 – 28: Was there a reason for using the non-standard solutions in the sondes at Gillot? If so this should be stated.

*The use of a non standard solution was a mistake.*

Page 9, Line 5: Insert "The" before "valid range".

*Corrected.*

Page 9, Line 7: delete "until", insert "near"

*Corrected.*

Page 9, 14-19. If a Morgane campaign curtain plot of the lidar retrieved aerosol scattering ratio is available, this would be a good place to insert that to go along with the discussion of the volcanic plume. At what altitude were the stratospheric intrusions located. Is this enhanced aerosol visible in the daily ASR plots? This should show up in the figure mentioned here.

*Enhanced aerosol loadings (likely coming from the Calbuco eruptions) were observed with the 532nm backscattered signal of the LIO3T (not shown) in these stratospheric air masses entering the troposphere above Reunion Island, which could have disturbed the ozone detection and quantification by the LIO3T, and consequently lower the agreement between LIO3T and ECC soundings during this period. These stratosphere-to-troposphere exchanges involving a volcanic plume above Reunion Island will be the subject of a dedicated study. This is now stated page 8 line 24-28.*

Page 10, Line 19: "co-located" should be changed to "compared"

*Corrected.*

Page 10, Line 23: "set a comparable" should be "set of comparable"

*Corrected.*

Page 12, Line 9: delete "to a NDACC labellisation" insert "for inclusion within NDACC"

*Corrected.*

**RESPONSE TO REVIEWER 2**

General Comments:

It should be reiterated throughout the text which lidar system (old vs. new) is being described. Although the subscript "_UR" is used to denote the old system, I still found several instances confusing as to which lidar data was presented.

*The structure of the article is changed to make it clearer along the text.*

Figure 1: Recommend adding ECC, LIO3T, and LIO3t_UR to the map after site location. Also altitude site altitude would be helpful.

*Figure 1 now indicates were the used instruments are located. Table 1 gives coordinates and altitude of the sites.*

Figure 2: The VR and uncertainty are effectively presented in Figure 6/7 for the new instrument. The information in Figure 2 could be simply stated in the text at various altitudes or added as a component to complement figure 6/7. Or conversely, bring in the new lidar VR/Uncertainty budget here as well to simultaneously show the absolute uncertainty as well as improvement to the system.

*VR and uncertainties are now provided for both systems on Figures 3 and 4, respectively.*

Figure 3: It would be useful to add the occurrences (with a different color) for the new system. It is stated later in the text and elaborated in Figure 14, but may make more sense to have towards the beginning to complement the old lidar system. Conversely, you could move all of the lidar/ECC discussion to the end (where Figure 14 currently exists).

*The article outline is modified and the ECC, LIO3TUR and LIO3T climatologies are given and discussed Section 5. Moreover, Figure 10 is now more consistent with Figure 11 and gives the monthly distribution of the number of profiles for ECC, LIO3TUR and LIO3T.*

Section 2.2 and Figure 4: Does the missing data (e.g. December above 10 km) in bottom panel Figure 4 correspond to lack of lidar coverage? Or the uncertainty of the measurement was larger than a certain threshold?

*We have only one December profile measured by LIO3T for the 01/2013-01/2016 period, and this profile ends up at 10km due to a misalignement of the lidar. It is stated page 11 lines 5-6.*

It would be helpful to recreate the occurrences in Figure 3 as the same layout as Figure 4 but monthly instead of yearly.

*Figure 10 is now more consistent with Figure 11 and gives the monthly distribution of the number of profiles for ECC, LIO3TUR and LIO3T.*

Generally the ECC shows higher ozone between 10-16 kms, regardless of time of year. Is this a known issue? Relevant to sonde reprocessing?

*The ECC climatology is now done with the reprocessed ECC database (Witte et al., 2017) and with a correction applied to take into account the non-standard ECC/solution pairing (cf. Section 4.1). As a result, these greater ozone concentrations between 10-16km previously pointed out by ECC do not appear anymore.*

and more importantly, are these differences within the uncertainty of the lidar measurement (with a set vertical resolution/temporal averaging)? A difference plot may highlight potential differences.

*The goal of this paper is not to validate LIO3TUR measurements, and ECC and LIO3T measurements are compared (for LIO3T validation) in Section 4.1 for both Gillot and Maïdo soundings with a better accuracy in the comparison than the one that would be obtained from a plot of the difference between two climatologies.*

Is the ECC climatology only from Gillot or combination of Gillot/Maido?

*Figure 11 only shows Gillot climatology.*

Figure 6: Are the differences observed between 6-8 km geophysical or based on the algorithm/processing? Is there a sounding for this observational period to compare with? I would assume the SNR to be largest from 6-8 kms and therefore the temporal averaging would generate less obvious differences. Is it possible there is a partial overlap or saturation correction that needs to be addressed?

*The differences are geophysical and unfortunately no sounding is available during the period. Anyway, we decided to remove this figure as it does not bring any information with respect to the scope of this article, which is not to analyse any case study.*

L25 – Remove "5The"

*Corrected.*

Figure 7: Are these uncertainties explicitly based on the retrievals performed in Figure 7 or are these average uncertainties over many nights. It seems odd that at 8 kms, there is a greater uncertainty from a 1 hr measurement than at 20 mins. Please clarify.

*These uncertainties are for the entire dataset. There was a mistake in the uncertainty calculation, due to the various valid ranges troughout the dataset. This is now corrected (Figure 4).*

*P8L23 – Is Witte et al., still in prep or under review?*

*Witte et al. is now in review. It is corrected page 7 line 24 and in the references.*

P9L9 – Should be McGee, not MacGee

*Corrected.*

Figure 8/9: There appears to be the largest difference between the lidar and ECC below 8 kms. Is this systematic for all 8 launches? Is this also apparent in the original lidar data?

*No, this not systematic for the 8 launches.*

Were these sondes launched from Gillot or Maido?

*Figure 5 shows comparison between LIO3T and ECC sondes launched from Maïdo, and Figure 6 between LIO3T and ECC sondes launched from Gillot.*

It is useful to have a profile of the number of comparisons to make discontinuities apparent.

*Right. Figures 5 and 6 now give the profile of the number of comparisons.*

If the potentially contaminated (e.g from aerosols) data were removed would the results improve?

*We did not make the exercise yet, but the use of the 532nm channel to detect and possibly discard aerosol layers from ozone retrieval is in our future plans.*

Eq 6, what is reference for the 200 in the numerator? Is this based on sample size?

*The 200 comes from the fact that we use the relative difference between two observations ("r"), using the mean value of these 2 observations to not considere anyone as the reference one. Equation 6 is now made more explicit.*

Figure 11: What's driving the difference in FTIR uncertainty from the beginning of the time series to the end? Is this the full uncertainty or partial uncertainty? Bottom panel ylabel can just be D.

*The full uncertainty is used here for the FTIR and the lower uncertainty shown in 2013 was a mistake. It is now corrected.*

Further suggestion #1: Instead of the cross histogram in figure 15, show a complete time series of all the data in the archive used to make this plot. Showing the reader the entire data set instead of the histogram will emphasize the observed variability in O3.

*This is a good suggestion. However, we chose not to present the LIO3T time serie because it is too sparse for the time being (84 profiles over 1084 days) to make it easily readible.*

Further suggestion #2: One very unique piece of information at the lidar site location is the ability to observe this enhanced O3 related to biomass burning. In the section describing the IASA comparisons, it may be helpful to show a world map of one day (or gridded month) where Maido is directly in the center of a CO transport plume, therefore highlighting the location and detection capabilities of the instruments.

*The interesting location of the site to observe biomass burning plumes is already documented in numerous previous studies using with ground-based (e.g. FTIR, lidar, ECC) and satellite (e.g. IASI, CALIPSO, MODIS) observations. Consequently, we do not think it is necessary to include such a world map in this paper.*

General comments :
This paper is indeed well in the scope of the journal, but needs to be improved before acceptance. The manuscript appears as not enough mature, missing frequently of accuracy with lack of definitions and arguments. Provide sections 3.2.1, 3.2.2 and 3.2.3 earlier in the manuscript because these informations should be available for the lidar when operated at l'Université de la Réunion.

*Sections 3.2.1, 3.2.2 and 3.2.3 are now provided earlier in the manuscript.*

Equation 6, correlated text and figures are unclear and I will not address any comments there, unfortunately a key point in your study. Improve Figures and to suppress the figures or table when not enough informative and avoid repetitions. Material should be valuable with a real effort and some work to reinforce the demonstration and the benefits/limits from this new instrumentation.

*Equation 6 is now clarified and "not-usefull-enough" Figures and Tables are removed.*

Specific comments (not extensive) :
Title should be revised discarding Part 1 which is not meaningful. I suggest : « Ozone profiles from a DIAL lidar at Maïdo Observatory (Reunion Island): instrumental description, instrumental performance, and result comparison with ozone external data set. »

*This suggestion is taken into account in the new title. However, "Part 1" is kept because this paper is the companion article of an upcoming one dealing with stratospheric ozone measurements in Reunion Island, which will be the "Part 2".*

Your paper is based on a DIAL instrument...but DIAL is nowhere defined as a Differencial Absorption Lidar. Please define this acronym at least once.

*The DIAL acronym is now defined in the abstract and in Section 2.1.*

This lidar technique is now well known and you refer to major technical points already published regarding your instrument (ex line 31 p3-line 1 p4) and data processing. Highlight what is new regarding LIO3T instrument and performance as compared to previous published papers.

*This work mainly aims to validate the LIO3T measurements. As, indeed, technical description of the instrument can be found in previous published papers, we now refer to these previous papers and only state the most important features of the system.*

Please discuss the uncertainty from your figure 2 versus figure 7 and explain why the results are similar/different with respect to altitude.

*There was a mistake in the uncertainty calculation due to the various valid ranges troughout the*

*lidars datasets. It is now corrected and results are more consistant.*

Homogenize your instrument labelling. Fix it once and make it consistent all along the text, including abstract, figures and tables. Along the text, you used : ozone lidar system (LIO3TUR), LIO3T, LIO3T O3, LIO3TUR, system, current system, current LIO3T system, LIO3T system, LIO3T lidar, LIO3T low and LIO3T high (this two latest are undefined elsewhere and used also in the abstract - unclear). For example I suggest to replace sentence starting line 10 page 3 by : "In the following, the lidar installed at the Université de la Réunion will be referred as LIO3TUR whereas the one installed at the Maïdo Observatory will be referred as LIO3T". Two labelling seems enough. If you need more, clearly explain.

*Instrument labelling is now consistent. However, we did not change line 10 page 3 because we do not want the reader to believe that both system (LIO3TUR and LIO3T) are currently operating (only the LIO3T – at the Maïdo Observatory – operates). Regarding the "LIO3T low" and "LIO3T high", they are not labels, they indicate if LIO3T is in average higher or lower than the ancillary dataset. It is now made more explicit.*

Revise and Clarify line 5 p2 : "Ozone is a major greenhouse gas in the upper troposphere and lower stratosphere". Insist on radiative forcing contribution and remove "major"...
Rephrase lines 14-15 p2 and justify "great interest" to document climate change.
Line 20 p2 : "To improve the operation of remote sensing instrument" remote sensing is imprecise. Do you refer to space-borne instrumentation?... and operation seems not appropriated. Replace "a 2160m-high atmospheric facility" by "a high atmospheric facility" to avoid repetition. You provide afterwards geographical coordinates and altitude.

*Introduction is now revised and clarified with respect to these comments.*

Figure 1 : Gillot in black is difficult to see and altitude would be here welcome (figure already published please provide reference).

*Figure 1 is changed and coordinates (including altitudes) are given for each site on Table 1.*

Additional general comment on altitude: mention once all altitudes in your manuscript will be provided in amsl and avoid to repeat latter.

*Done on page 6 line 14.*

Replace title section 2 by "Historical context of the lidar installed at the Université de la Réunion (LIO3TUR, 1998-2010)

*The article outline is changed and we now describe the LIO3TUR and LIO3T in the same Section (3.1).*

Replace title section 2.1 by "Instrumental description" or "Instrumental characteristics"

*Done.*

Line 7 p4 : First sentence is really too short...Is that your result? Could you explain the altitude range limitation? Don't forget your goal Line24 p3 is to provide data "over the entire tropospheric column...".

*This range validity is now justified by the overlap factor and signal-to-ratio (page 6 lines 13-14). Moreover, the "over the entire tropospheric column" sentence is now removed and replaced by "In 1998, an extension was installed to the existing system to perform ozone measurements in the free troposphere, including the upper troposphere" (page 5 line 22).*

Figure 2 : by "resolution" you mean "vertical resolution"...

*Yes, X label is now clarified.*

What is the criteria for uncertainty? It is defined lately for LIO3T.

*The same criteria apply for the LIO3TUR uncertainty and the Data processing sections are now provided before.*

It would be valuable in addition to have uncertainty expressed with the ozone unit provided. Explain why such a change in the uncertainty of LIO3TUR with respect to altitude. Please modify your X-axis (top and bottom) giving a more precise scaling (more minor ticks).

*Done.*

Line 9 p4 : "Temporal resolution... was chosen depending on atmospheric conditions"...Please justify what is your criteria.

*The atmospheric conditions mentioned here are the cloud free sky duration. It is now clarified in the text (page 6 line 16).*

Figure 3 and Figure 4 : please provide consistent figures. The number of profiles given per year is non adapted and consistent to support a monthly-averaged climatology. Note than < 10 profile per year is extremely low. Provide the monthly statistics for the ECC sondes, the LIO3TUR and LIO3T in a panel of your figure 4 , deep gaps within years could be specified in the text and you might suppress figure 3. Please replace figure 4 caption by "Monthly O3 climatology derived from ECC sondes over 2005-2015 at Gillot site between 0 and 19km (Top left panel), from LIO3TUR over 1998-2010 at Université de la Réunion campus site between X and XXkm (Top right panel) and from LIO3T over 2013-2015 at Maïdo Observatory between X and XXkm (Bottom panel).".

*Figure 10 now shows the monthly statistics for ECC, LIO3TUR and LIO3T and Figure 11 caption is*

*changed.*

Comparing the ECC sondes and LIO3TUR climatology, your figure 4 points out within 10-16km greater ozone concentrations from the ECC sondes...Explain, please.

*The ECC climatology is now done with the reprocessed ECC database (Witte et al., 2017) and with a correction applied to take into account the non-standard ECC/solution pairing (cf. Section 4.1). As a result, these greater ozone concentrations between 10-16km previously pointed out by ECC do not appear anymore.*

Replace title section 3 by "The lidar installed at the Maïdo Observatory (LIO3T, 2013-present)"

*As LIO3TUR and LIO3T are now presented in the same Section (3.1), this section does not exist anymore.*

Line 31 p4 : "Many instrument" is very imprecise.

*The sentence is changed.*

Line 32 p4 : "another lidar" is imprecise. Which one : the LI1200 ?

*Right. It is now clarified in the text.*

Replace 3.1 in title "current system". In this paragraph, a similar description has been provided in Baray et al. (2013) and citation is missing. Is there something new?

*Yes, description of the system can be found in Baray et al. (2013) and there is nothing new on this system since 2013. We now only state the main important features of the system.*

Suppress lines 18-20 from this paragraph, probably the right place would be in "conclusions and future plans".

*Done.*

Please consider Table 1 and Figure 5 and try to avoid repetition between both. Table 1 should be probably suppressed.

*This is right ; Table 1 is suppressed.*

Please modify in Figure 5 caption using "LIO3T instrumental schema" ... I expect you are referring

here precisely to LIO3T at Maïdo Observatory???

*Figure 2 caption is modified, and yes, we are refering to the LIO3T (at Maïdo Observatory).*

Line 23 p5 : "two backscattered lidar signal at two different wavelength" Check if correct.

*It is correct.*

Line 24 p5 : be more concise. I suggest "...wavelength, at 289nm (lon) where ozone is strongly absorbed and at 316nm (loff) where ozone absorption is weaker".

*This sentence is now more concise.*

Line 28 p5 to Line 7 p6 : please avoid repetitions. Fix once z as the altitude and l as the wavelength. Please provide the interfering gas.

*Repetitions are now avoided. Regarding the interfering gases, according to Leblanc et al. (2016), the interfering gases to consider in practice are NO2, SO2, and O2. NO2 and SO2 are negligible in most cases of tropospheric ozone retrieval, except in heavy volcanic aerosols loading conditions. The absorption by O2 should be considered if any of the detection wavelength is shorter than 294nm (which is the case here as we use the 289nm wavelength). However, in our case, we do not take into account any interfering gases for the time being. It is part of our future plans to include them in the "DIAL" code. This is now stated page 4 lines 3-8.*

Line 7 p6 : This last sentence introduces further 3.2.2, specify. Additionally replace "this" by the equation number to be precise.-

*Done.*

Replace 3.2.2 title I suggest to replace by Saturation, correction and consequences on the vertical resolution

*Done.*

Line 9 p6 : "Saturation" what is saturated and what is the cause and why below 7km.

*The saturation is defined as a difference between the number of photons received by the detector and the number of photons acquired. It is a non-linear phenomenon, depending on the dead time of the detector. In the LIO3T case, due to the detector sensitivity and the geometry of the instrument, we found that saturation occurs only below 7km. This is now clarified page 4 lines 10 and 16.*

Line 13 p6 : 'etc'. Rephrase the sentence"...type of filter such as....

*Done.*

Line 14 p6 : replace "described" by extensively detailed.

*Done.*

Line 15-16 p6 : "we decide"...replace by we found and rephrase but add (not shown)

*Done.*

Line 18 p6 : "the resolution rises" Modify expression. Compare the vertical resolution to figure 2 and explain.

*The sentence is corrected and the vertical resolutions are presented in Figure 3 and Section 3.2.*

Figure 6 : Modify the resolution lines using dashed lines and modify caption on the figure in order to discriminate ozone profiles and the vertical resolution given with respect to integration time. In the figure caption, origin of data (LIO3T) is missing.

*Figure 6 is suppressed it does not bring any information with respect to the scope of this article, which is not to analyse any case study.*

Line 26 p6 : Please check here "to 5The"

*Done.*

Line 9-10 p7 : "new tropospheric ozone version..." the details on cross-sections and uncertainty you provide is not much informative and information seems different from line 25-26 p6.

*This sentence is suppressed (not very usefull indeed).*

Line 11-17 p7 : not at the right place, better place could be in conclusions and future plans

*Done.*

Line 25-26 p7 : explain why only at night.

*Done page 6 line 13.*

Explain why such integration time changes with respect to the data you compare to.

*Done page 6 line 27 and page 8 line 6.*

Line 28-30 p7. Please rephrase this sentence. "Notably" is imprecise, how much and where, but note the consistency within ~9-11,5km and explain.

*We removed this figure as it does not bring any information with respect to the scope of this article, which is not to analyse any case study.*

Line 4 p8 : "goes" replace by varies

*Done.*

Figure 7 : Why LIO3TUR uncertainty increases by a factor of 3 at least above 13km whilst the one of LIO3T decreases above 14km whatever the integration time by a factor of 2 at least.

*There was a mistake in the uncertainty calculation due to the various valid ranges troughout the lidars datasets. It is now corrected and results are more consistant.*

Revise section 4 title. Comparison is really too short here. What pieces are you comparing.

*Section 4 title is now "Comparisons with ancillary data".*

Line 9 p8 'validate' Do you compare, evaluate or validate?...Keep constant and take care when using validation concept. I would say here you evaluate...

*One of the main goals of this paper is to validate the LIO3T measurements by comparing them to ECC soundings (which are commonly used for this purpose). We consider that this work is a validation of the LIO3T measurements.*

Line 13p 8 add IASI after space-borne and replace instrument by data.

*Done.*

Line 14-16: This paragraph is absolutely obscure... I don't understand at all what you are dealing with and description of the equation terms is hard to follow. Be clear, concise, avoid repetition and be simple, that will help. I can't go further here... Explain clearly how can D be negative and give a clear definition for D ??? What is LIO3Tn? What is MCDn? What is D by the end?

*We are very sorry that this "D" parameter confused you and hamper your reading of the manuscript. "D" definition is now clarified (page 7 equations 7 and 8). "r" is the relative difference*

*between two observations, using the mean value of these 2 observations to not considere anyone as the reference one. "D" is the mean over the entire dataset of the absolute values of "r". We use the absolute values of "r" because positive and negative values can balance each other resulting in an erroneous "D" (which can not be negative indeed - X label of right panel plots of Figures 5 and 6 is now "r" instead of "D").*

Line 17 p8 add sondes after ECC.

*Done*.

Line 23-28 p8 : For that reason I think you should use only 'evaluate' and not 'validate'...as mentioned above.

*Following the work of Johnson et al. (2002, 2016) intercomparing various KI and buffer solutions, we found that this ENSCI/0.5% full buffer solution tends to overestimate the amount of ozone by 1.7% in the troposphere. Consequently, an adpated correction was applied on the ECC profiles during this period. This is stated page 7 line 28 and page 8 line 1. Having corrected these ECC observations, we consider we can use them to validate the LIO3T measurements.*

Line 9 p9 "SO2 loading too strong" imprecise, please provide informations on the amount.

*This aerosol enhancement is clearly visible on the 355nm channels of the stratospheric ozone and LI1200 lidars, and on the 532nm channel of the LIO3T (not shown), and back trajectories together with CALIOP observations (on board CALIPSO - not shown) show that the detected plume comes from the Calbuco volcano. Consequently, although we do not have any information on the corresponding aerosol and SO2 amount, we consider as a wise assumption that, in the layer where this volcanic plume lies (i.e. between 17 and 22km), the SO2 and aerosols loading is too strong to allow a correct O3 retrieval. The study of this volcanic plume crossing in the south-western Indian Ocean will be the subject of dedicated articles. These explanations are added page 8 lines 12-17.*

Line 11 p9 : "mean D" see above comment and remove LIO3Tlow which is undefined and that I do not understand. Clear definition is mandatory.

*D only gives an absolute relative difference between two datasets, without indicating which one is greater than the other. "LIO3T low" means that the LIO3T dataset gives lower values than the ancillary dataset. It is now made more explicit.*

Line 15 p9: "Enhanced" by how much and provide a reference.

*These sentences warn the reader about specific conditions that occurred during the MORGANE campaign, and introduce an ongoing work about these stratospheric intrusions above Reunion Island involving the Calbuco volcanic plume. Consequently, no reference is available. This paragraph is modified according to this answer.*

Line18-19 p9: Not at the right place, move to conclusions and future plans.

*Done.*

Figure 8 and 9 : I am not able to make any comment at the moment. Please use different line thickness for the mean and standard deviation in the figures. Text of the figure caption might be improved. Use same Yaxis altitude range in both figures.

*Figures 5 and 6 are now improved with respect to these comments.*

Line 21-22 p9 : Replace 3PM by 15:00:00 LT, 7PM by 19:00:00 and 1AM by 01:00:00 LT

*Done.*

Line 25 p9 : replace "goes"

*Done.*

Line 29-30 p9: rephrase this sentence, unclear.

*This sentence is suppressed because the ENSCI/0.5% full buffer solution effect is now corrected.*

Line 2 p10 : This FTIR spectrometer should be added to the list of instruments operating at Maïdo Observatory and provided in introduction (line 26-27).

*We talk in the introduction only about the "lidar systems that are permanently deployed and routinely operated at the Maïdo Observatory". As of this writing, 30 instruments are installed in the Maïdo Observatory, and this paper does not aim to list all of them. However, Figure 1 is changed and the FTIR appears on it, together with its location.*

Line 19-20 p10 : improve text...both instrument are operating at the same place so their measurements are colocated...What has to be pointed out is the time window you consider for the comparison.

*This sentence is made clearer.*

Add "Thus, for the LIO3T comparison with FTIR, 114 LIO3T profiles are available".

*114 FTIR-LIO3T pairs are available, because a single LIO3T profile can be linked to several FTIR measurements (within the time window).*

Line 21 p10 : just mention the selected LIO3T profiles are regridded consistently to the FTIR.

*Done.*

Line 22 p10 How many FTIR data are averaged within 24h.

*It depends on the measurement conditions (the FTIR only operates in clear daily skies).*

Line 23-31 p10: Very very hard to follow...

*Sorry about that !*

Line 3 p11: replace "time series" by "available over the 01/2013-01/2015 period."

*Done.*

Figure 11 : The caption is imprecise and bottom plot Y-axis text is not consistent with equation 6. Is it D???

*Y label is now "r".*

Moreover, I found 12 symbols on this bottom plot ???

*That is right. Number is changed page 10 line 4.*

Figure 12 : I wonder if figure 10 and 12 should be gathered...

*Good idea, Figure 10 and 12 are now gathered into Figure 7.*

Provide minor ticks for the month on the Xaxis plots

*Done.*

Line 18 p11 : suppress " hereabove for the comparison with the ground-based FTIR observations" and replace by "in section 4.2.1".

*Done.*

Figure 13: refer to figure 11 comment

*Done.*

Line 27 p11 : I don't clearly see a seasonal cycle, I see an O3 increase particularly in 2013 and 2016 suggesting an impact of biomass burning...

*This sentence is corrected (page 10 line 32).*

Section 5 : Title is data set and times series... You are dealing here with data from Intensive period of observations during campaigns... I think it makes the difference with material in previous sections, if I well understood. Please provide a more explicit title. This section is lengthy.

*In Section 4, we were dealing with comparison with ancillary data coming from both intensive (Maïdo ECC) and routine (Gillot ECC, FTIR and IASI) periods of observations. In Section 5, we present the full datasets and the resulting climatologies. Section 5 title is changed to "Dataset and climatologies". Moreover, this section is now shortened.*

Line 30 p11 – line 3 p12 : You provide LIO3T results in this section...Thus other lidar and details not used further are out of the scope of your study. Thus suppress. This text is very long and the Figure 14 do not provide more informations with respect to the text and is difficult to read. Shorten and improve text.

*Text is shorten and Figure 14 is removed because Figure 10 now shows the monthly total number of LIO3T profiles.*

Figure 15 is not much informative and text seems to repeat what is given in Line19-21 p7.

*Figure 15 is suppressed.*

Figure 4 : Please specify in the caption of bottom panel what the LIO3T climatology includes (data routinely performed and from intensive period of observations ???).

*It is now specified in the caption that the plot includes data routinely performed and from intensive period of observations*

Figure 16 : please provide these informations (4 numbers) on Figure 17 and suppress Figure 16.

*Figure 16 is suppressed and information are given in Figure 12 caption.*

Line 16-20 p12 : bring in the light what are the benefits from your new lidar...For sure a better description from the upper-troposphere/lower-stratosphere than when located at UR.

*Exactly. It is now stated in the text (page 12 lines 18-20).*

Line 21-24 p12 : valuable comments and for such reasons I encourage authors to carefully and

rigorously revised the manuscript. Take care to the ECC caveats already mentioned.

*OK.*

Figure 17 : I recommend to add informations on the seasonal sampling frequency with respect to altitude and this should be done here with an additional panel.

*Done on Figure 12 (right panel).*

Section 6 : bring more in the light the benefits provided from LIO3T...if monthly climatology from ECC is equivalent and to LIO3TUR and LIO3T (i.e. range of values and seasonal patterns). Could you reinforce you study here... Your goal was to describe the whole tropospheric column with LIO3T... What is your conclusion?

*The move of this lidar from the Université de la Réunion campus site up to the Maïdo observatory allows it to document the UT/LS region and to follow stratospheric and tropospheric intrusions with relevant vertical and time resolutions together with a reasonable uncertainty (1.5km, 20min and 14%, respectively, at 18km). This tropospheric ozone data set covering the tropical free troposphere and UT/LS of a sparsely documented region (South Western Indian Ocean) constitutes an extremely valuable resource for the validation of satellite tropospheric ozone retrievals, analysis of the ozone variability and sources, dynamics analysis of case studies, and for long term atmospheric monitoring. This is now stated page 12 lines 18-23.*

Line 15-20 p12 : specify altitude range here.

*Done.*

Line 29-32 p12 : suggestion : A DIAL tropospheric ozone lidar was operating on the Université de la Réunion campus site from 1998 to 2010, providing 427 ozone profiles.

*OK.*

Note that this information on 427 profiles was not mentioned before. Same remark for LIO3T profiles.

*It was mentioned page 4 line 11 for the LIO3TUR and page 12 line 4 for LIO3T.*

Replace "familly" by network.

*Done.*

Line 9 p13 : "we found a 7.7% D between"...revised with the D definition.

*Done.*

[revised manuscript text omitted]

---

## Author Response (AR2)

*Dear Dr Stiller,*

*We thank the reviewer for the comments and suggestions. Please find below a point by point answer to the reviewer's comments on the last version of the manuscript.*

Comments on the title:

Despite my recommendation in AMTD to withdraw from the title "Part 1", which was not meaningful, authors kept "Part 1" in their ATM manuscript submitted. Their justification is : "this paper is the companion article of an upcoming one dealing with stratospheric ozone measurements in Reunion Island, which will be the "Part 2".". As far as I know I haven't seen a reference to this paper in the text nor any introduction whilst they mention futures studies on aerosols p 6 line 11 in the manuscript. Has it been even more submitted as a companion paper? From my point of view, the link seems weak and the justification is not acceptable. Final decision is under the responsibility of the editor and editorial board.

*The reviewer is right: the companion paper should have been submitted ~4 months ago, and, as it was delayed by the first author, it makes no sense anymore to keep the "Part 1" in our title. We removed it and the new title is: "Tropospheric ozone profiles by DIAL at Maïdo Observatory (Reunion Island): system description, instrumental performance, and result comparison with ozone external data set".*

General comments:

This paper, indeed well in the scope of the journal, does not meet the standard criteria to be published in AMT. I am not addressing here a whole list of suggestions/corrections. The main motivations for rejection are :

1/ Despite my previous recommendations the manuscript still remains with incomplete, undefined, imprecise or incorrect definitions or equations (in particular in section 2.2 and 2.3, and in section 4 equation 7 and 8, see brief details further). An equation in the text should help the reader to understand precisely what calculation has been done which is not the case here. DIAL lidar ozone data retrieval, a quite complex technique to provide accurate results, is extensively documented in previous published papers. Here the synthesis is unclear and explanations appear not under control. Missing Page 3, line 22 the following reference "Harris at al. 1998" ("N. Harris, R.D. Hudson, and C. Phillips, Editors, WMO, SPARC/IOC Ozone Profile Trend Assessment, WMO Global Ozone Research and Monitoring Project - Report #43, Geneva, 1998.").

*Sections 2.2 and 2.3 were rewritten and the suggested reference was added.*

2/ Regarding the instrument itself and data processing (sections 2), no instrumental improvement is described in this manuscript, it has been detailed in previous papers. Additionally, no improvement in the technical ozone retrieval from DIAL despite authors planned to consider in the future the impact of aerosols and interfering gases. That would be for sure a valuable improvement to motivate a publication in AMT as you mentioned the impact of a Chilean volcano eruption on your data set.

*The main goals of this paper are to evaluate the LIO3T performances and to validate the LIO3T measurements with ancillary data. There is no publication dealing with neither the validation of*

*LIO3T observations nor the evaluation of its performances (the same remark applies for LIO3T_UR - except comparison with 2 ECC soundings in Baray et al., 1999). We believe it is fundamental to publish such a work to show to the community that the LIO3T measurements are trustworthy.*

I do not agree with your "saturation" definition p 4 line 10. Saturation from my point of view occurs when the photo counting system cannot handle the high intensities received by the detector. Saturation can't be low, it is saturated or not. What is "desaturation", then? In Eq 4, what is k, what is n, where is the altitude dependence? I want to know when your calculations are time or altitude or wavelength dependent (a general comment on your equations).

*Following the Hamamatsu Handbook for Photomultiplier tubes (Edition 3a, page 196), the saturation is defined "as the phenomenon in which the amount of output signal is no longer proportional to the incident light intensity". So saturation is not binary, it can be low or high. However, we changed the definition in the manuscript to be more precise (page 4 line 10).*
*The vertical resolution part (page 4 lines 17-22) was rewritten to take into account the reviewer's comments.*

Section 2.3, absolutely unclear (eq 5: associated error whilst you use standard deviation and signal to noise ratio; unclear from lines 10 to 15 including eq 6) should be replaced. When and where occurs the error propagation (line 12 p5)? It is not explained. What is xi, etc? Please be accurate in your description.

*This section 2.3 was rewritten.*

To conclude, by considering the previous published publications, a change in the altitude location of the instrument is not a sufficient motivation to support a publication in AMT and given the text clarity and improvements, the paper appears as not mature from the technical point of view.

*We do not agree with such a conclusion. We present the system as it is for the time being, exposing its actual performances, limitations and identified axes of improvement. We show through intercomparison with external ozone dataset that the performed measurements are trustworthy.*

3/ Regarding the LIO3T performances (section 3), they were evaluated considering 427 LIO3TUR profiles against 84 LIO3T profiles. Why the lidar LIO3TUR operates at night to increase SNR (signal to noise ratio, undefined) is not justified. The "overlap factor" is not defined.

*We added a reference to justify that operating at night increases the signal to noise ratio, and we provide a definition of the overlap factor (page 5 line 31).*

Comments on Figure 3 should be addressed straightforward on the LIO3T vertical resolution improvement/deterioration as compared to LIO3TUR. Line 15 p 6 is not consistent regarding your Figure 3 and 4.

*We first use Figures 3 and 4 to describe the vertical resolution and uncertainty of LIO3T_UR measurements, before decribing those of LIO3T measurements using the same Figures. Line 15 p 6 (now page 6 line 2) is then consistent with Figures 3 and 4. However, we did not explain the differences in the signal filtering between LIO3T_UR and LIO3T. It is now done Section 2.2 (page 4*

*lines 17-22). Moreover, a comment is now addressed on the observed difference between the LIO3T_UR and LIO3T vertical resolutions page 6 line 19.*

Line 32 p 6 do not specify the instrument except if you refer to figure 4 and you did that also in lines 16 -19 page 6.

*This is right. It is now specified that we refer to the LIO3T.*

My feeling is that section 3.2 needs to be reorganize/rewritten to better distinguish LIO3TUR from LIO3T informations.

*In this Section 3.2, we already comment LIO3T_UR and LIO3T performances separately.*

Benefits from this instrument altitude change vs vertical resolution and uncertainties are not brought into light.

*The main benefit of the instrument altitude change (document the UT/LS region with relevant vertical and time resolutions together with a reasonable uncertainty) is already stated in the Conclusion (page 12 lines 8-13). However, we state it again in this Section (page 6 lines 31-32).*

I found the message in your concluding remarks not enough straightforward and clear (p7 line4-6 – SNR, uncertainty, detection noise with respect to ozone variability in UT/LS).

*The sentences are modified to make them clearer (page 6 lines 26-30).*

4/ Regarding the comparison with ancillary data.

A/ ECC sondes : Number of profiles or partial columns included in the comparison is missing in p7, line 8-12.

*Numbers of comparison pairs are now given page 7 lines 1-6.*

They are only 8 for ECC sondes launched (at night time???) at Maïdo Observatory in collocation and time-coincidence within ± ?? hours (be accurate).

*It is now specified that these 8 ECC sondes were launched in time-coincidence with the lidar shooting page 7 lines 25 and 28-29.*

They are 37 for ECC at Gilot day time launched compared to full night LIO3T with a ± ?? hours delay (be accurate)…

*It is already in the text (page 8 lines 16-18): "Figure 6 shows the comparison between the SHADOZ/NDACC Gillot routine ECC soundings and LIO3T profiles. As the first ones are performed during daytime (usually around 15:00:00 LT) and the last ones during night time (between 19:00:00 and 01:00:00 LT), ECC soundings are taken into consideration when performed one day before or after a LIO3T profile acquisition."*

Introduce your data set precisely once in section 4 and do not repeat later.

*Done.*

What is N, M, MCD and rn? It should be a very simple calculation and text is unclear. Consider fig 5 in Gaudel et al., 2015 as an example, it is simple and clear. I only agree with your r (which is rn in the text, what n stands for?) if you specify that is altitude dependent (z). But now how to interpret D? You can't conclude on a bias high or low with D? You just provide a value and conclude it is in good agreement…

*N, M, MCD, n, and rn are already defined in the text (page 7 lines 7-11).*
*Figure 5 in Gaudel et al. (2015) deals with ECC vs lidar partial columns seasonal comparison over 5-year periods. Figure 5 (right column) gives the mean difference ECC minus lidar for each of the seasons and each of the 5-year periods. Gaudel et al. do not explain how they calculated the mean difference between datasets.*
*Let's take an example : let's compare 2 datasets A and B each containing 3 elements. When calculating the difference for each data pairs (A(1)-B(1), A(2)-B(2), A(3)-B(3)), let's assume one finds the following values (in ppb or whatever) : -4, +6, -2. If one calculates the mean difference ((-4+6-2)/3), one would find zero, which could mean that the datasets are in perfect agreement (which is obviously not true). More generally, mean differences calculated this way tend to be underestimated.*
*Considering the mean of the absolute value of the differences ((4+6+2)/3), one would find 4, which gives a far better view on the (dis)agreement between datasets. Nevertheless, it does not provide any information on the sign of the relative difference, which should be given separatly (in this particular example there is no sign as the result is zero).*
*We try in this comparison exercises to be the more honest as possible by considering ways of calculation (for both $r_n$ and D) that do not decrease the results (i.e. that do not increase artificially the agreement between datasets).*
*However, the Reviewer is right on one point: on right panel of Figures 5 and 6, the X-axis label should be "mean(r)" instead of "r". It is now corrected.*

Further in the text authors specified that some of the 8 compared cases (how many exactly?) are sampled in a context of high aerosols due to volcano Calbuco eruption (Chile, not located). The full date and time page 8 lines 3 and 4 are missing even if table 2 is providing dates but it is impossible to learn which is which. My feeling is that you have here a valuable material to make progress in your retrieval technique.

*Dates of the profiles that are impacted by the Calbuco eruption are now italicized in Table 2 and the Calbuco volcano is located.*

I am not sure the line 9-18 are essential to interpret your results and figures at the moment because you do not take into account the aerosols in the DIAL retrieval and you do not show the data. It is just to introduce further studies??? If not essential, withdraw and also the text repeated in lines 23-28 p 8 or if essential improve the retrieval.

*Lines 9-18 are important to explain why the used profiles are limited up at 17km ; lines 23-28 were removed.*

In the Figure 5 caption, you use "accuracy" which is undefined.

*"Accuracy" was replaced by "precision" in the Figure 5 caption and in the text (page 7 line 1) which refers to Smit et al. (2007).*

Now if we consider the 37 ECC sondes launched at Gillot, what is the benefit of this second comparison to external data because launched within less time coincidence and collocation, i.e. leading to increase the probability to sample different air masses. You conclude on a greater D (9.4% as compared to 6.8% for Maïdo). The justification for a comparison with a second set of ECC sondes is not provided... What are the benefits to your demonstration?

*We try here to compare the LIO3T measurements with all the available external O3 data we have, and the Gillot ECC soundings are a very valuable database to do so.*

Finally is LIO3T bias high or low as compared to LIO3TUR? You can't use D to conclude on this last point. Need clarifications.

*We do not understand this question: LIO3T_UR and LIO3T measurements are not (and can not be) compared to each other.*

B: FTIR ground based, IASI and LIO3T : 12 partial columns are taken in the intercomparison for the ground based FTIR and 39 for IASI, both instruments over 3 years. D is 11.8 and 11.3% respectively, that is greater than with ECC sondes by almost a factor of 2. A long text to describe instrument and data retrieval compared to really poor conclusions and interpretations (p10 lines 4-11).

*The comparison between lidar and FTIR is not straightforward, and needs explanations. The conclusions of these intercomparisons are not poor, they are simple.*

5/ Regarding the data set and climatologies (section 5) : Fig 11 is a valuable result and I suggest to add an ECC climatology over 2013-2015, the LIO3T period considered.

*We do not see the value of adding a 2013-2015 ECC climatology ; in a LIO3T validation point of view : comparing it with the 2013-2015 LIO3T climatology would not improve the reliability of the ECC vs LIO3T comparison made with time-collocated ECC soundings ; in a geophysical point of view : the goal here is not to compare tropospheric ozone climatologies (by lidar or ECC) at different periods.*

That is interesting but the discussion is poor : I haven't seen much improvement in the quality of LIO3T climatology as compared to ECC sondes and even more in UT/LS. What benefits are you expecting from this instrumental altitude change as compared to routinely ECC sondes measurements...

*The goal of this paper is not to convince the readers that lidars are valuable instruments for the atmosphere observation. It is well known that, as compared to routine ECC soundings, one of the main interests of lidar measurements is to allow the following of short time-scale processes, such as stratosphere-to-troposphere exchanges. And the benefits of such an instrumental altitude change is already stated in the conclusion page 12 lines 8-13 (and page 6 lines 31-32).*

Please compare your results to what Gaudel et al, 2015 have published in figure 4. What can you conclude? Are your results consistent? For all those reasons, I found the paper not mature.

*There must be a mistake here : Figure 4 in Gaudel et al. (2015) shows a map of the average of NOx emissions (from 1991 to 2010), which is not in the scope of this paper.*

Minor details (among the lot of remaining errors…) :
- A very poor and confusing English style. In addition, relationship between sentences and paragraphs are deficient.
- Along the text, correct use of higher (an altitude is higher) and greater (an amount is greater). "Ancillary data", is very imprecise as compared to "O3 external data set".

*Corrected.*

FINAL CONCLUSIONS AND RECOMMENDATIONS : I don't accept this manuscript at this stage for a publication in AMT, even in the scope of the journal. I recommend a new submission accordingly to my comments with exact definitions, to increase the number of LIO3T measurements (including 2016 profiles at least) and to improve the retrieval technique by taking into account the aerosols. In order to provide a substantial, consistent and condensed technical overview of the DIAL technique at Maïdo Observatory, I suggest to add part 2 dedicated to stratospheric ozone. Therefore the new submitted paper with substantial material could be valuable for further ozone studies which use the DIAL lidar at Maïdo Observatory as expected by authors and co-authors.

*Once again, we do not agree with such a conclusion. Definitions and data processing details are provided or make reference to previous published works. We present the system as it is for the time being, exposing its actual performances, limitations and identified axes of improvement. We show through reliable intercomparison exercises with external ozone dataset that the performed measurements are trustworthy. DIAL stratospheric ozone measurements performed at Reunion Island need a dedicated publication. And, despite the fact that aerosols are not taken into account in the retrieval scheme for the time being, we did show that measurements agree well with external datasets.*

[revised manuscript text omitted]